# Age-related accumulation of de novo mitochondrial mutations in mammalian oocytes and somatic tissues

**Barbara Arbeithuber**[1], **James Hester**[2], **Marzia A. Cremona**[3¤a], **Nicholas Stoler**[4], **Arslan Zaidi**[1¤b], **Bonnie Higgins**[1], **Kate Anthony**[1], **Francesca Chiaromonte**[3,5], **Francisco J. Diaz**[2], **Kateryna D. Makova**[1]*

**1** Department of Biology, Pennsylvania State University, University Park, Pennsylvania, United States of America, **2** Department of Animal Science, Pennsylvania State University, University Park, Pennsylvania, United States of America, **3** Department of Statistics, Pennsylvania State University, University Park, Pennsylvania, United States of America, **4** Department of Biochemistry and Molecular Biology, Pennsylvania State University, University Park, Pennsylvania, United States of America, **5** EMbeDS, Sant'Anna School of Advanced Studies, Pisa, Italy

¤a Current address: Department of Operations and Decision Systems, Université Laval, Québec, Canada
¤b Current address: Department of Genetics, Perelman School of Medicine, University of Pennsylvania, Philadelphia, Pennsylvania, United States of America
* kdm16@psu.edu

**Data Availability Statement:** The sequencing reads are available at SRA under accession PRJNA563921. The data were analyzed in R using packages listed in the Methods. Computer code for

## Abstract

Mutations create genetic variation for other evolutionary forces to operate on and cause numerous genetic diseases. Nevertheless, how de novo mutations arise remains poorly understood. Progress in the area is hindered by the fact that error rates of conventional sequencing technologies (1 in 100 or 1,000 base pairs) are several orders of magnitude higher than de novo mutation rates (1 in 10,000,000 or 100,000,000 base pairs per generation). Moreover, previous analyses of germline de novo mutations examined pedigrees (and not germ cells) and thus were likely affected by selection. Here, we applied highly accurate duplex sequencing to detect low-frequency, de novo mutations in mitochondrial DNA (mtDNA) directly from oocytes and from somatic tissues (brain and muscle) of 36 mice from two independent pedigrees. We found mtDNA mutation frequencies 2- to 3-fold higher in 10-month-old than in 1-month-old mice, demonstrating mutation accumulation during the period of only 9 mo. Mutation frequencies and patterns differed between germline and somatic tissues and among mtDNA regions, suggestive of distinct mutagenesis mechanisms. Additionally, we discovered a more pronounced genetic drift of mitochondrial genetic variants in the germline of older versus younger mice, arguing for mtDNA turnover during oocyte meiotic arrest. Our study deciphered for the first time the intricacies of germline de novo mutagenesis using duplex sequencing directly in oocytes, which provided unprecedented resolution and minimized selection effects present in pedigree studies. Moreover, our work provides important information about the origins and accumulation of mutations with aging/maturation and has implications for delayed reproduction in modern human societies. Furthermore, the duplex sequencing method we optimized for single cells opens avenues for investigating low-frequency mutations in other studies.

selection analysis is in S2–S4 Notes. All raw data for the information depicted in figures are available at: https://github.com/makovalab-psu/mouse-duplexSeq.

**Funding:** This project was supported by a grant from NIH (R01GM116044) for KDM and a Schrödinger Fellowship from the Austrian Science Fund (FWF) for BA: J-4096. Additional funding was provided by the Office of Science Engagement, Eberly College of Sciences, The Huck Institute of Life Sciences and the Institute for Computational and Data Sciences at Penn State, as well as, in part, under grants from the Pennsylvania Department of Health using Tobacco Settlement and CURE Funds. The department specifically disclaims any responsibility for any analyses, responsibility or conclusions. The funders had no role in study design, data collection and analysis, decision to publish, or preparation of the manuscript.

**Competing interests:** The authors have declared that no competing interests exist.

**Abbreviations:** CpG, 5′-cytosine-phosphate-guanine-3′; CpT, 5′-cytosine-phosphate-thymine-3′; CSB, conserved sequence box; DCS, duplex consensus sequence; ETAS, extended termination-associated sequence; hN/hS, number of nonsynonymous mutations per nonsynonymous site/number of synonymous mutations per synonymous site; H-strand, heavy strand; ID, identifier; L-strand, light strand; MAF, minor allele frequency; mtDNA, mitochondrial DNA; NGS, next generation sequencing; Numt, regions of nuclear DNA that are homologous to mtDNA; Polg, DNA polymerase subunit gamma; SSCS, single-strand consensus sequence.

## Introduction

Mutations generate genetic variation, making evolution possible. Substantial progress has been made in our understanding of mutation rates and patterns because of the application of next generation sequencing (NGS) to interspecies genomic alignments (reviewed in [1]), familial trios [2–9], mutation accumulation lines (reviewed in [10]), and mutator animal models [11,12]. Yet the error rates of available NGS technologies (e.g., $10^{-3}$ to $10^{-2}$ for Illumina [13] and even higher for others) are several orders of magnitude higher than the mutation rates themselves (e.g., $3.5–5.4 \times 10^{-9}$ and $7.69 \times 10^{-8}$ mutations per base pair per generation in mouse germline and brain, respectively, for nuclear DNA [14,15]). As a result, conventional NGS cannot directly provide accurate answers to several critical questions related to mutations —for instance, how de novo mutations arise and whether their frequency increases with age.

Multiple lines of evidence suggest that, in humans, nuclear germline mutations increase in frequency with both paternal and maternal age [2–5,7,9], a conclusion that is highly relevant to modern societies in which delayed reproduction has become common. A paternal age effect for nuclear germline mutations was recently also observed in mice [15]. However, existing studies infer germline mutation frequency increases indirectly from the analysis of familial trios; thus, the phenomenon has never been directly shown to occur in germ cells. Without evidence gleaned directly from oocytes, the increased frequency of de novo genetic variants in the offspring could be due to decreased selective constraints, rather than to enhanced mutagenesis, in the germline of older mothers. We currently lack such direct evidence from oocytes not only for nuclear DNA but also for the mitochondrial genome.

Mitochondria, the powerhouses of the cell, have their own genomes. Multicopy mitochondrial DNA (mtDNA) genomes—e.g., present in thousands of copies in brain and skeletal muscle and in approximately 200,000 copies in mature oocytes [16–20]—are transmitted in mammals through the maternal lineage. Studying how mutations in mtDNA accumulate with age is critical because of their association with numerous human genetic diseases (reviewed in [21]) as well as with aging phenotypes [22], including Parkinson's and Alzheimer's diseases [23,24]. The substitution rate of mtDNA is approximately an order of magnitude higher than that of the nuclear DNA [25–27], providing a higher resolution for detecting changes in mutation frequency compared with nuclear mutations. In human somatic tissues, mtDNA mutation frequency increases with age [26–30]; however, whether the same occurs in mice remains contradictory ([11,31–33], reviewed in [34,35]).

Recent analyses of human pedigrees suggested age-related accumulation of mtDNA mutations in the human female germline, leading to more de novo mutations in children born to older mothers [26,36]. Thus, de novo mtDNA mutations likely accumulate in oocytes under meiotic arrest. A direct examination of mtDNA variants in oocytes has been challenging. Most studies to date focused on a handful of mtDNA sites [37–42]. The few that did examine the whole mtDNA sequence [35,43–45] used sequencing methods with high error rates and, thus, potentially unreliable detection of de novo mutations. In practice, we still do not know definitively whether the frequency of de novo mtDNA mutations increases with age in mammalian oocytes.

To investigate age-related accumulation of mtDNA mutations in mammalian somatic and germ cells, we optimized and applied a highly accurate duplex sequencing method [28,46,47] to mtDNA in brain, skeletal muscle, and oocytes of mothers and pups belonging to two mouse pedigrees. This approach allows one to accurately estimate mutation frequencies as low as $<10^{-7}$ [28,46,47]. Using it, we were able, for the first time, to directly measure the frequency of de novo germline mutations across the whole mitochondrial genome and to demonstrate their accumulation in mothers compared with pups. Additionally, we examined changes in allele

frequencies of inherited mtDNA mutations between tissues and generations and the dependence of these changes on age, thus illuminating population genetic processes that govern mtDNA evolution in mammalian somatic and germline tissues.

## Results

### Duplex sequencing of two mouse pedigrees and mutation detection

To study the accumulation of mtDNA mutations in somatic and germline tissues, we generated high-quality full-length mtDNA sequences from brain, skeletal muscle (henceforth referred to as muscle), and oocytes of CD-1 mice from two independent two-generation pedigrees (Fig 1). A total of eight mothers and 28 of their pups (19 females and nine males) were included in the study. At the time of sample collection, the age of the mothers was approximately 10 months, and the age of the pups was approximately 20 days (exact ages are reported in S1 Table). mtDNA from brain and muscle was studied for every animal. Additionally, for 25 females, we studied oocyte mtDNA; this included mtDNA from 1–8 single oocytes per animal for 21 females (a total of 92 single oocytes) and mtDNA from (usually) 4–47 pooled oocytes per animal for 24 females (a total of 24 oocyte pools).

To sequence mtDNA, we used the highly accurate duplex sequencing technique [46,48]. By barcoding double-stranded sequencing templates for each strand before amplification during library preparation, duplex sequencing separates true DNA variants from sequencing errors [46,48]. This is achieved by first forming consensus sequences for reads originating from each of the two template strands separately, i.e., single-strand consensus sequences (SSCSs), which still contain some PCR errors and artifacts resulting from DNA lesions, followed by forming a consensus of the two single-strand consensuses, i.e., duplex consensus sequence (DCS), in

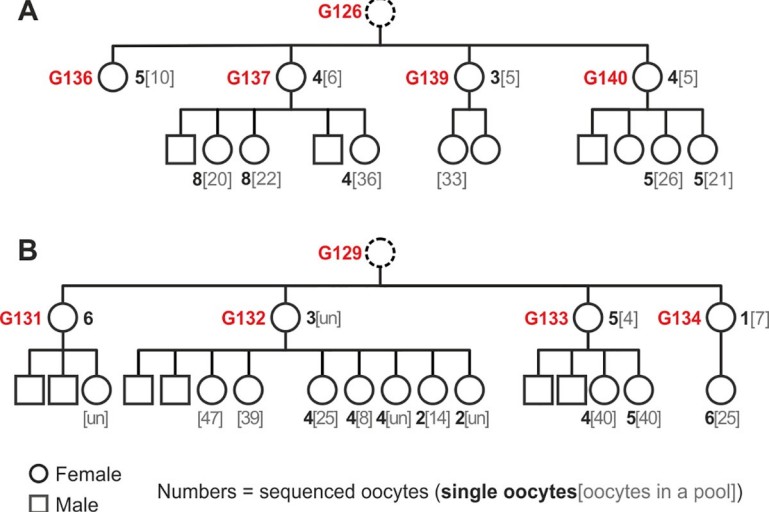

**Fig 1. Two mouse pedigrees included in the study.** At the time of sample collection, the age of the mothers was approximately 10 mo, whereas the age of the pups was approximately 20 d. Red numbers are individual IDs. Black numbers in bold indicate the number of single oocytes sequenced per individual, and gray numbers in parentheses indicate the number of oocytes included in a pool per individual (only one oocyte pool was included per individual). (A) Pedigree with four mothers (sisters: G136, G137, G139, and G140) and 11 pups. This pedigree is further referred to as G126, based on the ID of the grandmother (who was not included in the study). For simplicity, mouse G136 is also referred to as a mother despite her not having pups. Pups of G137 were born in two litters (S1 Table), separated by a large gap on the figure. (B) Pedigree with four mothers (sisters: G131–G134) and 17 pups. This pedigree is further referred to as G129. Pups of G132 were born in two litters (S1 Table). ID, identifier; un, unknown number of oocytes included in a pool.

which these errors and artifacts can be eliminated (S1 Fig). The overall sample preparation and sequencing workflow are summarized in S2 Fig. Prior to library preparation, the samples were enriched for mtDNA to minimize the contribution from regions of nuclear DNA that are homologous to mtDNA (Numts) [49,50] (S3A Fig). Samples were sequenced on the Illumina HiSeq 2500 platform using 250-nt paired-end reads to a median DCS depth of 2,050×, 1,986×, 133×, and 932× for brain, muscle, single oocytes, and oocyte pools, respectively (S3B Fig, S2 Table), with a rather uniform DCS sequencing depth per site (S3C Fig). Sequencing reads were analyzed with the Du Novo pipeline [51,52] to form consensus sequences (S1 Fig), which were mapped to the mouse mtDNA reference genome and used to call variants (see Methods for details).

Although we observed 28 inherited variants (i.e., variants segregating in a pedigree) in this data set (see section Inherited heteroplasmies), our main interest was in the analysis of putative tissue-specific de novo mutations. We initially defined them as variants (nucleotide substitutions) present in one tissue of an animal but absent from the other tissues. Considering the average frequency of these mutations, we expect some de novo mutations to occur at the same site in two or three samples just by chance (S1 Note); therefore, we included such mutations in our analysis (we excluded early somatic mutations [$n$ = 54, S3 Table, S4 Fig]—mutations found in both somatic tissues but absent from oocytes of the same individual). Mutations occurring more than three times at a site ($n$ = 148) represent either mutation hot spots or variants segregating at very low frequencies in a pedigree; however, we cannot differentiate between these two possibilities. Thus, conservatively, we excluded these mutations from our analysis. In total, we found 325, 390, 218, and 38 putative tissue-specific de novo mutations in brain, muscle, single oocytes, and oocyte pools of mothers, respectively, and 465, 424, 78, and 77 mutations in the same tissues of pups. Considering only mutations occurring in a single sample, or considering all observed mutations, did not change our conclusions qualitatively (S1 Note). The majority of putative tissue-specific de novo mutations (1,885 out of a total of

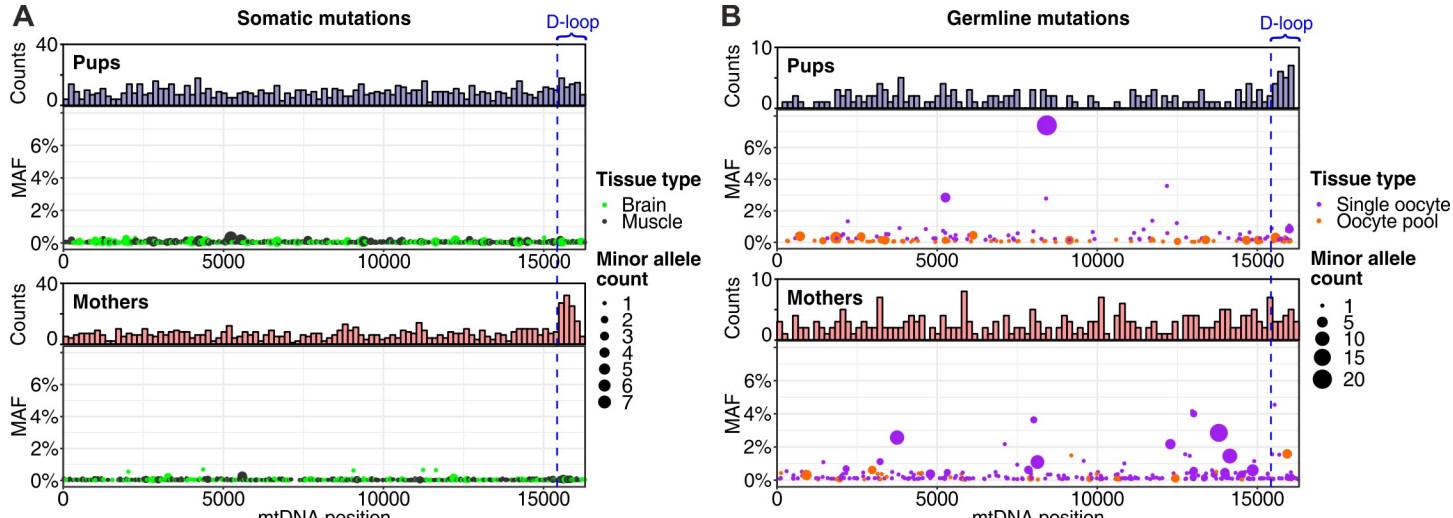

**Fig 2. Mutations along the mtDNA molecule in pups and mothers.** Distributions of (A) somatic and (B) germline mutations along the mitochondrial genome. Dot size represents the number of molecules (i.e., the number of DCSs) in which a mutation was observed. The majority of mutations were found only in a single molecule; therefore, the shown MAF might be inaccurate in some samples. Among the mutations observed in at least two DCSs (for which MAF can be estimated more accurately), only 11 had MAF >1%—they all occurred in the germline (two mutations in single oocytes of pups, eight mutations in single oocytes of mothers, and one mutation in oocyte pool of a mother; S3 Table). The dashed blue line marks the beginning of the D-loop region. The raw data for the information depicted in this figure are available at https://github.com/makovalab-psu/mouse-duplexSeq. DCS, duplex consensus sequence; MAF, minor allele frequency; mtDNA, mitochondrial DNA.

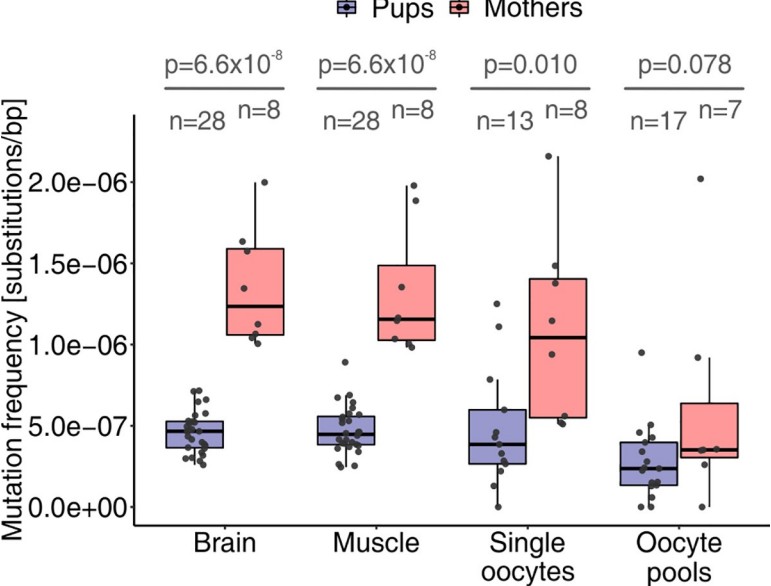

**Fig 3. Nucleotide substitutions accumulate with age.** Higher median mutation frequencies were observed in brain, muscle, single oocytes, and oocyte pools of mothers compared with those of pups. Permutation test *p*-values (one-sided test based on medians, corrected for multiple testing) are shown with the respective sample sizes (*n*). In the calculation of mutation frequencies for single oocytes, the numbers of mutations and sequenced nucleotides were combined across all oocytes analyzed for a mouse. Considering only samples sequenced at a DCS depth ≥100× does not change the results qualitatively (S5B Fig). The raw data for the information depicted in this figure are available at https://github.com/makovalab-psu/mouse-duplexSeq. DCS, duplex consensus sequence.

2,015) was found in a single mtDNA molecule (Fig 2A and 2B), suggesting that they have low frequencies.

## De novo mutations

**Increase in the frequency of somatic and germline mutations in older versus younger mice.** Our analysis of mice belonging to the two different age groups—approximately 10-mo-old mothers versus approximately 20-d-old pups—demonstrated that the frequency of de novo mtDNA mutations increases with age in both somatic tissues and germ cells (Fig 3). For each of the two somatic tissues analyzed, the mutation frequency was significantly higher in the mothers than in the pups. Indeed, we observed a 2.6-fold increase in mutation frequency in brain (median frequency among mothers: $1.2 \times 10^{-6}$ nucleotide substitutions/bp; median frequency among pups: $4.7 \times 10^{-7}$; one-sided permutation test $p = 6.6 \times 10^{-8}$; one-sided tests were used when comparing mothers and pups for quantities that are reasonably expected to be larger in mothers; *p*-values throughout the manuscript were corrected for multiple testing; see Methods) and a 2.6-fold increase in mutation frequency in muscle (mothers: $1.2 \times 10^{-6}$ nucleotide substitutions/bp; pups: $4.5 \times 10^{-7}$ nucleotide substitutions/bp; one-sided permutation test $p = 6.6 \times 10^{-8}$) in mothers compared with pups. For oocytes—either single oocytes or oocyte pools—the mutation frequency was also higher in mothers than in pups (though only significant for single oocytes). When analyzing oocyte pools, we observed a 1.5-fold increase in median frequency in mothers compared with pups (mothers: $3.5 \times 10^{-7}$ nucleotide substitutions/bp; pups: $2.4 \times 10^{-7}$ nucleotide substitutions/bp; one-sided permutation test $p = 0.078$), whereas for single oocytes the increase was 2.7-fold (mothers: $1.0 \times 10^{-6}$ nucleotide substitutions/bp; pups: $3.9 \times 10^{-7}$ nucleotide substitutions/bp; one-sided permutation test $p = 0.010$;

we combined the data from single oocytes per mouse in order to minimize the contribution of oocytes sequenced at low depth to the estimated mutation frequency; mutation frequencies in single oocytes are shown in S5A Fig).

Whereas in the preceding paragraph we considered mutation frequencies in each tissue of each individual separately, in the analysis below we aggregated mutations in each tissue across mothers and separately across pups to gain statistical power. We also aggregated data for single oocytes and oocyte pools because they represent the same tissue (separate data for single oocytes and oocyte pools are shown in S4 and S5 Tables). After such aggregations, our data still confirmed significant increases in somatic and germline mtDNA mutation frequencies in the mothers compared with the pups: we found significant increases of 2.8-fold, 2.7-fold, and 2.3-fold in mothers compared with pups for brain, muscle, and oocytes, respectively (Fisher's exact test $p = 4.0 \times 10^{-41}$, $p = 1.4 \times 10^{-41}$, and $p = 2.7 \times 10^{-16}$, respectively; S5C Fig, S4 Table).

**De novo mutation frequency and its increase with age vary along the mtDNA molecule.** Across all tissues, and for either mothers or pups, aggregated mutation frequencies were notably higher in the D-loop than in protein-coding, tRNA, or rRNA sequences. These differences were significant only in brain and muscle of mothers and in oocytes of pups (Fig 4A, S5 Table, S6 and S7 Figs). In pups, de novo mutations in the germline exhibited a particular preference for the D-loop versus all other compartments taken together (significant 3.1-fold-higher frequency in oocytes; histogram in the upper part of Fig 2B, Fig 4A; see S4 Table for mutation frequencies and Fisher's exact test p-values). This was also the case, but less markedly, in somatic tissues of pups (significant 1.6-fold-higher frequency in brain, $p = 0.012$, but nonsignificant 1.3-fold-higher frequency in muscle, $p = 0.216$, Fisher's exact tests; Figs 2A and 4A; S4 Table).

With aging, somatic mutations accumulated primarily in the D-loop, whereas germline mutations accumulated more uniformly along the mtDNA molecule (histograms in the lower parts of Fig 2A and 2B). Indeed, for somatic tissues, whereas all mtDNA functional

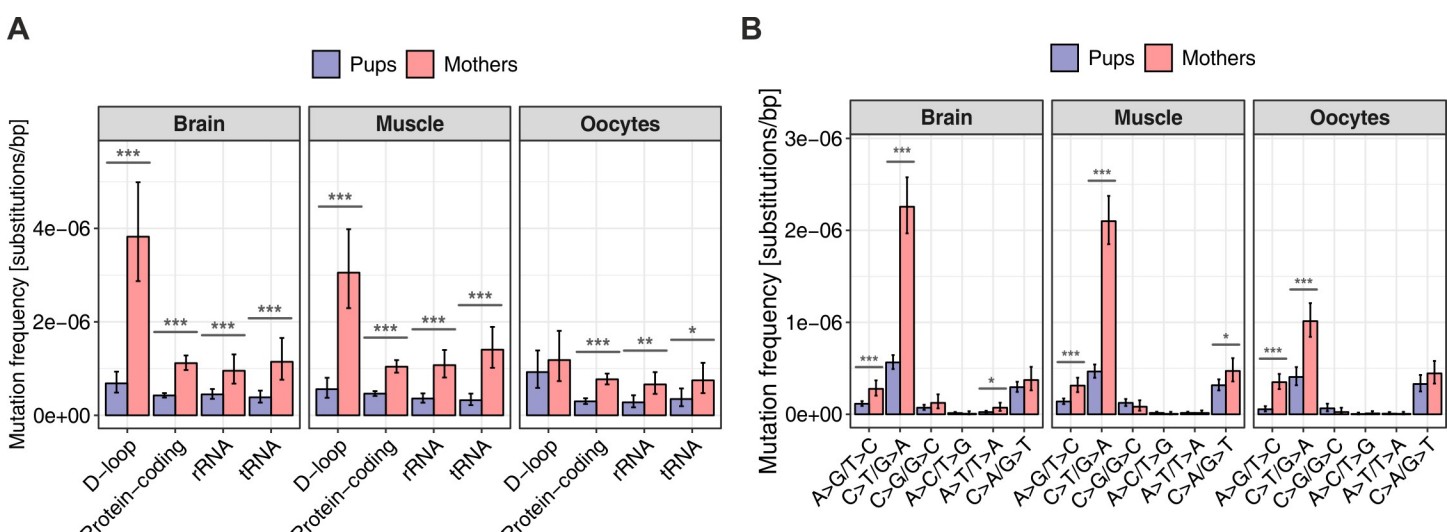

**Fig 4. Mutation patterns in pups and mothers.** (A) Tissue-specific mutation frequencies (computed as total number of tissue-specific mutations divided by the product of mtDNA region length and mean DCS depth) for pups and mothers in different mtDNA regions: D-loop (877 bp), protein-coding (11,331 bp), tRNA (1,499 bp), and rRNA (2,536 bp) sequences. Mutation frequencies for noncoding sequences outside of the D-loop (57 bp) are shown in S6 Table. (B) Tissue-specific frequencies of different mutation types for pups and mothers. C>T/G>A mutations separated by CpG and non-CpG sites are shown in S8 Fig. Mutation frequency bars are shown with 95% Poisson confidence intervals. Differences between mothers and pups in each category were tested using the Fisher's exact test; stars indicate p-values (*$p < 0.05$, **$p < 0.01$, ***$p < 0.001$). "Oocytes" comprise single oocytes and oocyte pools in both (A) and (B). The raw data for the information depicted in this figure are available at https://github.com/makovalab-psu/mouse-duplexSeq. CpG, 5'-cytosine-phosphate-guanine-3'; DCS, duplex consensus sequence; mtDNA, mitochondrial DNA.

compartments exhibited significant increases in mutation frequency in mothers versus pups (Fig 4A), these increases were larger for the D-loop (significant 5.6-fold and 5.5-fold increases in mothers versus pups in brain and muscle, respectively; see S4 Table for mutation frequencies and Fisher's exact test $p$-values) than for mtDNA outside of the D-loop (significant 2.6-fold and 2.5-fold increases in brain and muscle, respectively; S4 Table). Germline mutations did not show such a pronounced preference for the D-loop with age as that observed in somatic tissues (Fig 4A). In fact, germline mutations located in the D-loop did not have a significant increase in their frequency in mothers versus pups (1.3-fold increase, Fisher's exact test $p$ = 0.549), in contrast to germline mutations located in protein-coding, rRNA, and tRNA sequences, which displayed significant increases (2.6-fold, 2.4-fold, and 2.1-fold increases, respectively; Fig 4A, S4 Table).

In a complementary analysis of observed versus expected numbers of de novo mutations based on the length of mtDNA functional compartments, we also found the D-loop to harbor a higher number of mutations than expected by chance for all three tissues analyzed and for both mothers and pups (S6 Table). Consistent with our results in Fig 2A and 2B, the differences between observed and expected ratios in the D-loop were particularly striking for somatic tissues in mothers ($p$ = $2.2 \times 10^{-13}$ in brain and $p$ = $2.9 \times 10^{-10}$ in muscle; one-sided binomial test) and oocytes in pups ($p$ = $1.0 \times 10^{-5}$), showing an enrichment of mutations in the D-loop. De novo mutations were always significantly underrepresented in coding regions (protein-coding, tRNA, and rRNA considered together) for all three tissues analyzed and for both mothers and pups—again, particularly for somatic tissues in mothers ($p$ = $7.5 \times 10^{-13}$ in brain and $p$ = $2.7 \times 10^{-9}$ in muscle) and oocytes in pups ($p$ = $9.1 \times 10^{-6}$). Even though protein-coding regions constitute the majority of coding regions (11,331 bp out of a total of 15,366 bp), we found them to be strongly depleted of de novo mutations in mothers' brain and muscle ($p$ = $6.6 \times 10^{-3}$ and $4.2 \times 10^{-4}$, respectively) and in pups' oocytes ($p$ = 0.045). This pattern most likely reflects high mutation rates in the D-loop for these tissues and not negative selection against de novo mutations in protein-coding regions. Indeed, we observed that protein-coding regions exhibited nonsynonymous-to-synonymous rate ratios (number of nonsynonymous mutations per nonsynonymous site/number of synonymous mutations per synonymous site [hN/hS], [53]) of 1.30, 2.17, and 1.32 in brain, muscle, and oocytes of mothers, respectively, and 1.59, 1.19, and 1.69 in the same tissues of pups, all within the range of neutral expectations (S2 Note). Additionally, with the exception of brain tissue in pups, conservation (measured by phastCons scores) was not significantly lower for mutated sites relative to sites that were homoplasmic (S3 Note). Thus, there is little evidence to support the role of negative selection in shaping the observed distribution of mutations in mouse mtDNA in the tissues and age groups analyzed. Within the D-loop, we did not see a significant depletion of mutations in regulatory regions (extended termination-associated sequence [ETAS]1, ETAS2, conserved sequence box [CSB]1, CSB2, and CSB3) [27] compared with nonregulatory ones (S4 Note), again providing no evidence of selection.

**Age accumulation of different mutation types in germline and somatic tissues.** Our results suggest that during aging/maturation, transitions accumulate more than transversions, particularly in somatic tissues (S7 Table). When we compared mutation frequencies between mothers and pups, we observed significant 3.6-fold, 3.7-fold, and 3.2-fold increases in the frequency of transitions in brain, muscle, and oocytes (Fisher's exact test; $p$ = $4.9 \times 10^{-47}$, $3.4 \times 10^{-53}$, and $6.1 \times 10^{-21}$, respectively) but only 1.5-fold, 1.2-fold, and 1.2-fold increases in the frequency of transversions ($p$ = 0.019, 0.255, and 0.328, respectively). The transition-to-transversion ratio was 4.3, 4.5, and 3.3 in mothers' brain, muscle, and oocytes, respectively, as compared with only 1.8, 1.4, and 1.2 in the same tissues of pups.

Although all types of transitions displayed significant increases in mutation frequencies in mothers compared with pups (Fig 4B; see S7 Table for mutation frequencies and Fisher's exact test *p*-values), for A>G/T>C transitions, the age-related frequency increase was strongest in the germline, whereas for C>T/G>A transitions, the frequency increase was strongest in somatic tissues. The frequency of A>G/T>C transitions increased with age as much as 6.4-fold in the germline but only 2.4-fold in brain and 2.2-fold in muscle (Fig 4B). In contrast, the frequency of C>T/G>A transitions increased only 2.5-fold in oocytes but as much as 4.0-fold in brain and 4.5-fold in muscle. The frequency of C>T/G>A transitions was higher at 5′-cytosine-phosphate-guanine-3′ (CpG) sites than at non-CpG sites across all tissues for mothers (1.9-, 1.9-, and 2.4-fold difference for brain, muscle, and oocytes, respectively) and for pups (1.8-, 1.2-, and 1.4-fold difference for brain, muscle, and oocytes, respectively). The elevation in transition frequencies at CpG sites was significant for all maternal tissues but only for brain for pups (see S7 Table for mutation frequencies and Fisher's exact test *p*-values). Thus, spontaneous deamination of methylated cytosines [54] might be contributing to mouse mtDNA mutagenesis, particularly for older animals. However, no significant differences were observed between mothers and pups for any trinucleotide context individually (S9 Fig).

Although the patterns of age-related accumulation inside and outside the D-loop were similar for most substitution types and tissues (S8 Table, S10 Fig), C>T/G>A and A>G/T>C transitions in oocytes displayed a difference. Their frequency increased in mothers as compared with pups significantly, 2.7- and 7.1-fold, outside the D-loop but not significantly, only 1.3- and 3.3-fold, in the D-loop. The latter observation might be explained by the small number of these mutations in the D-loop in our data set and should be confirmed in other studies using larger sample sizes.

**Similar patterns of strand bias in younger versus older animals.** We detected strand bias, i.e., an asymmetric distribution of complementary mutation types between the light strand (L-strand; containing more cytosines than guanines) and the heavy strand (H-strand) for several mtDNA mutation types and in somatic and germline tissues of our mice—similar to what was shown previously for mtDNA in human brain [28]. The patterns of strand bias were largely similar between younger and older animals. The mouse reference mtDNA genome represents the L-strand; therefore, all mutation types below are shown with respect to that strand (with duplex sequencing, mutations were measured on both DNA strands but only reported with respect to the reference sequence). However, some of the actual mutations might have occurred on the H-strand instead. For instance, an inferred G>A mutation on the L-strand might have in reality been a C>T mutation on the H-strand. If there is no strand bias and if the nucleotide composition of the two strands is the same, we expect similar numbers of reciprocal mutations (e.g., C>T and G>A, and other mutation types combined in each category in Fig 4B), using either strand as reference.

We observed a significantly higher frequency of G>A than C>T mutations for brain and muscle at non-CpG sites and for brain at CpG sites in both mothers and pups, for oocytes at CpG and non-CpG sites in mothers, and for oocytes at non-CpG sites in pups (S11 Fig, S9 Table). For instance, significant 5.2-fold- and 7.9-fold-higher frequencies of G>A than of C>T mutations were observed in mothers' and pups' brain (see S9 Table for mutation frequencies and Fisher's exact test *p*-values). We also observed a significantly higher frequency of T>C than A>G mutations in brain and muscle of mothers (5.7-fold- and 2.5-fold-higher frequency) and in brain and muscle of pups (2.7-fold- and 2.9-fold-higher frequency). The D-loop, unlike other mtDNA compartments taken together, did not show a significant strand bias for transitions (S10 Table, S12 Fig), but this observation could in part be due to a relatively small number of mutations of different types in the D-loop.

**Table 1. Mixed-effects binomial logistic regression for mutation frequency at mtDNA (see S11 Table for details).**

| Predictor | Partial pseudo-$R^2$ (%) | Chi-squared test $p$-value |
|---|---|---|
| Mutation type | 12.69 | $3.06 \times 10^{-251}$ |
| Age category (mothers or pups) | 4.99 | $1.67 \times 10^{-13}$ |
| mtDNA region (D-loop or outside of D-loop) | 3.43 | $1.01 \times 10^{-21}$ |
| Tissue | 2.77 | $1.11 \times 10^{-18}$ |

Model conditional pseudo-$R^2$: 23.53% (overall predictive power). Model marginal pseudo-$R^2$: 22.82% (predictive power of the fixed effects).

Abbreviation: mtDNA, mitochondrial DNA

**A joint model explaining variation in mtDNA mutation frequencies.** To evaluate the joint effect and the relative contribution of different factors to mtDNA mutations, we fitted a generalized mixed-effects linear model using a binomial family and a logit link (see Methods). Mutation frequency was the response; the total number of nucleotides (in DCSs) was used as weight; and age category (mothers or pups), tissue (brain, muscle, single oocytes, or oocyte pools), mtDNA compartment (D-loop versus outside of D-loop), and mutation type (transversions, A>G/T>C transitions, and C>T/G>A transitions) were used as fixed-effect categorical predictors (Table 1). The data on single oocytes were combined per individual, and individual identifier (ID) was included as a random effect to account for potential differences among individuals. In all, the model captured a substantial portion of the variability observed in mutation frequency (conditional pseudo-$R^2$ = 23.53%). All four categorical predictors were significant (Table 1, see S11 Table for odds ratios). The strongest predictor was mutation type (partial pseudo-$R^2$ = 12.69%). The model confirmed a higher mutation frequency of C>T/G>A transitions versus transversions (Fig 2D); however, the frequency of A>G/T>C transitions was not significantly different from the frequency of all transversions combined (S11 Table). The second strongest predictor was age category (mother or pup; partial pseudo-$R^2$ = 4.99%). The model corroborated a significantly higher mutation frequency in mothers versus pups (Fig 3, S11 Table). mtDNA compartment was the third strongest predictor (partial pseudo-$R^2$ = 3.43%), and the model confirmed a significantly higher mutation frequency in the D-loop versus other compartments (Fig 4A, S11 Table). Finally, tissue was a significant but weaker predictor (partial pseudo-$R^2$ = 2.77%). The model confirmed lower mutation frequencies in oocytes than in brain and indicated no significant differences in mutation frequencies between brain and muscle (S11 Table), confirming our observations throughout the manuscript. Individual ID was a significant random effect ($p = 2.1 \times 10^{-7}$); however, its role in the model was minor. Indeed, taken together, the fixed-effect predictors captured almost as much of the variability in mutation frequencies as the whole model (marginal pseudo-$R^2$ = 22.82%, compared with the conditional pseudo-$R^2$ = 23.53%), which also accounts for the random effect. A separate model that included two-way interactions between mutation type and each of the other three variables in addition to the four fixed-effects categorical variables listed above captured a similar amount of variability in mutation frequency (S12 Table) as did the model without interactions (Tables 1 and S11).

## Inherited heteroplasmies

**Variability in allele frequency at inherited heteroplasmies.** Although our main focus was on de novo mutations, which we had a unique opportunity to evaluate with the duplex sequencing protocol, the two mouse pedigrees also provided us with data to study inherited heteroplasmies. In our data set, we found 28 inherited heteroplasmic sites, which we defined as

variants present either in both somatic and germline tissues of an individual or in two generations of a pedigree (S13 Table, S13 Fig). Among them, 17 heteroplasmies were present in both somatic tissues and oocytes of one or several pups but were absent from their mothers (S13 Table); these mutations likely originated in the germline of the mothers and were inherited by the pup(s). Seven heteroplasmies were shared by a mother and some of her pups and thus likely originated in the mother (or in an earlier generation). Four heteroplasmies were shared by several mothers and their pups in a pedigree and thus were likely also present in the grandmother. Median allele frequencies (S14 Fig, S14 Table) were the lowest for heteroplasmies restricted to pups (median frequency across samples of 0.35%), intermediate for heteroplasmies shared by a mother and her pups (median frequency across samples of 0.59% and 2.68% in mothers and pups, respectively), and highest for heteroplasmies shared by several mothers (median frequency across samples of 9.01%). This illustrates an increase in allele frequencies between new mutations and mutations segregating across several generations or a higher probability of high-frequency heteroplasmies to be inherited [27,36].

**More pronounced age-related genetic drift in germline than somatic tissues.** Analyzing all 28 inherited heteroplasmic sites, we were able to make inferences about the amount of genetic drift for mouse mtDNA and found that, in somatic tissues, it is limited and increases only slightly with age. Heteroplasmic allele frequencies correlated tightly between the two somatic tissues of pups ($R^2 = 0.977$; Fig 5A). In comparison, mothers (approximately 9 mo older than pups) exhibited only a slightly lower correlation ($R^2 = 0.947$; Fig 5B; this conclusion did not change when we subsampled pups to have a sample size equal to the sample size for mothers, S15 Table).

Our results suggest a more pronounced genetic drift and a larger increase in the amount of drift with age for mtDNA in mouse germline than somatic tissues. When comparing allele frequencies at these 28 sites between somatic tissues and oocytes (single oocytes and not oocyte pools were used to obtain approximately equal numbers of oocytes for individual mothers and pups), correlations in both pups and mothers ($R^2 = 0.871$ and $R^2 = 0.592$, respectively; Fig 5C and 5D) were weaker than those observed between their somatic tissues ($R^2 = 0.977$ and $R^2 = 0.947$, respectively; Fig 5A and 5B). This finding can be explained by the bottleneck experienced by mouse mtDNA in the germline [37]. In addition, the correlation observed for mothers ($R^2 = 0.592$) was substantially lower than that observed for pups ($R^2 = 0.871$), suggesting stronger drift for oocytes of older animals, which is likely due to the additional turnover of mtDNA during meiotic arrest (transfer of mitochondria from neighboring cyst cells during oogenesis might also contribute to the increased drift [55]). These conclusions did not change when we subsampled the data to include the same number of mothers and pups, limited the analysis to samples (mothers or pups) with more than two oocytes, or computed correlations for individual oocytes (S15 Table, S15 Fig). We found additional evidence that genetic drift increases in aging oocytes but much less so in aging somatic tissues: the variance in minor allele frequency (MAF), normalized by $p(1 - p)$, where $p$ is the average allele frequency among single oocytes or between somatic tissues, was significantly higher in mothers than in pups when considering oocytes but not when considering somatic tissues (one-sided permutation test $p$-values of 0.047 and 0.478, respectively; S16 Fig).

**Estimating the effective germline bottleneck for mouse mtDNA.** Our data are consistent with a germline bottleneck operating on mouse mtDNA, corroborating previous studies (previous estimates in mice are approximately 200 segregating mtDNA units) [37]. We next estimated the size of the effective germline bottleneck for mouse mtDNA—the size required to explain observed genetic drift, which might differ from the actual number of transmitted mtDNA molecules because of their potential segregation in groups (reviewed in [56]) and/or the transfer of mitochondria from cyst cells [55]. To estimate the effective germline bottleneck

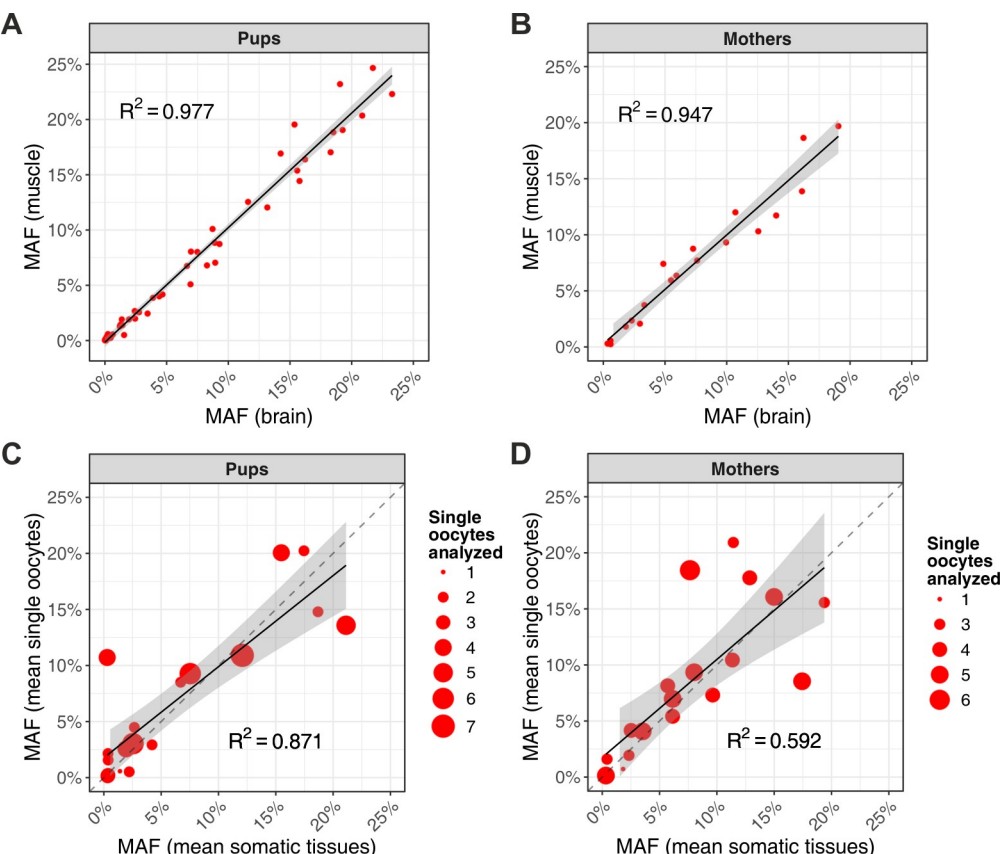

**Fig 5. Correlations of allele frequencies for inherited heteroplasmies in somatic tissues and oocytes.** (A) MAF of heteroplasmies in brain versus muscle for pups. (B) MAF of heteroplasmies in brain versus muscle for mothers. (C) Mean MAF of heteroplasmies in single oocytes versus somatic tissues (averaged between brain and muscle) for pups. (D) Mean MAF of heteroplasmies in single oocytes versus somatic tissues (averaged between brain and muscle) for mothers. In (C-D), single oocytes (not oocyte pools) were used to obtain approximately equal numbers of oocytes for individual mothers and pups. Dot size indicates the number of sampled oocytes for each pup or for each mother. The gray bands on all plots represent confidence intervals around the regression line, and the dashed lines are the 1:1 relationship. The raw data for the information depicted in this figure are available at https://github.com/makovalab-psu/mouse-duplexSeq. MAF, minor allele frequency.

size, we used the population genetics approach described in [57–59]. Applying the approach in [57–59] to the data on allele frequency shifts at 28 inherited heteroplasmies, we compared allele frequencies between somatic tissues of pups and their single oocytes (at 19 heteroplasmic sites with one to seven oocytes per animal), between somatic tissues of mothers and their single oocytes (at 19 heteroplasmic sites with one to six oocytes per animal), and between somatic tissues of mothers and somatic tissues of pups (at 16 heteroplasmic sites with one to nine pups per mother; S16 Table). The resulting effective bottleneck size was estimated to be 84.8 segregating mtDNA units (95% bootstrap CI 0.0640–462), 53.6 segregating units (95% bootstrap CI 1.65–472), and 59.1 segregating units (95% bootstrap CI 8.26–315), respectively (S16 Table). The effective bottleneck size obtained from comparisons including mothers is smaller than that obtained from comparisons between somatic and germline tissues of pups, suggesting that additional drift occurs with aging. Therefore, the estimate based on pups' data is closer to the initial germline bottleneck size.

## Discussion

Utilizing a highly accurate duplex sequencing method, we were able to study in detail both age-related accumulation of putative de novo mutations and changes in allele frequencies for inherited variants in mouse mtDNA. Our results demonstrate that the frequency of putative de novo mtDNA mutations significantly increases (approximately 2- to 3-fold) as mice age from approximately 20 d to approximately 10 mo, i.e., over a period of only 9 mo, and in both somatic tissues and germ cells.

### Power of duplex sequencing

Duplex sequencing greatly reduces errors resulting from DNA damage and amplification in NGS experiments (error rate: $<10^{-7}$) [46]. As a result, we could reliably detect low-frequency de novo mutations occurring at frequencies below 1%, which is often used as a cutoff in mtDNA studies [26,33]. Additionally, because PCR occurs after template tagging in duplex sequencing, our estimates of allele frequencies for both de novo and inherited mtDNA variants are expected to be more accurate than in studies using conventional NGS.

### Innovative application of duplex sequencing to single oocytes

Although duplex sequencing was successfully applied to de novo mtDNA mutation detection in somatic tissues in previous studies [28,46,60], ours is the first study to apply this method to single oocytes. Duplex sequencing of single cells is particularly challenging because of their small DNA quantity. The amount of mtDNA in a single oocyte (<5 pg) is several orders of magnitude lower than the minimal amount of DNA required for library preparation in a published duplex sequencing protocol (approximately 100 ng) [48]. DNA amplification prior to library preparation is not an option for duplex sequencing because, to distinguish true variants from errors/artifacts, the original double-stranded state of each template molecule needs to be preserved until duplex adapters are attached. To accommodate the low mtDNA amount in single oocytes, we had to implement several modifications to the library preparation protocol (see Methods). The procedure we thus optimized for application to single cells is an important outcome of this study; we expect it to become a useful resource for other researchers interested in investigating de novo mutations at a low DNA amount or single-cell level, e.g., during cancer or due to exposure to environmental pollutants or radiation.

### Mutation accumulation in the germline

The application of duplex sequencing to single oocytes and oocyte pools allowed us to analyze putative de novo mtDNA mutations directly in germ cells (i.e., without the need to observe them in the next generation) with a resolution sufficient to capture frequency increase over a period of only 9 mo. Previous studies in humans are contradictory and either suggested no increase in the number of mutations in single oocytes with ovarian aging [35] or an increase in the number of de novo germline mutations with maternal age at conception [26,36] (the latter studies were based on the analysis of new heteroplasmies in the offspring, thus providing only indirect measurements of germline mutation frequencies). Studies of this phenomenon in mice have been scarce. Ma and colleagues [33] found an increase in the number of inferred germline mutations (defined as variants present in multiple somatic tissues of an individual) in older versus younger wild-type mice. However, this observation could be explained by an increase in MAF of inherited heteroplasmies in older animals (crossing the detection threshold). Our results suggest that in wild-type mice, de novo mutations are difficult to observe with the MAF = 2% detection threshold used in [33]; thus, many low-frequency mutations

were missed in their analysis. Furthermore, many of the germline mutations identified in their study might represent inherited heteroplasmies. In contrast, our study unequivocally shows an age-related increase in de novo mtDNA mutation frequency in mouse female germ cells. However, we note that because we measured mutations only at two time points (approximately 20 d and approximately 10 mo), the observed increase may be linked not just to age directly but also to other aging-related factors, such as parity or sexual maturity.

## Mutation accumulation in somatic tissues

Ours is also the first application of duplex sequencing to demonstrate age-related accumulation of somatic mtDNA mutations in mice. Our results agree with those by Vermulst and colleagues [31], who, using a random mutation capture assay and comparing brain and heart of young versus old mice, found a significant increase in the frequency of point mutations at two targeted mtDNA positions (in the rRNA region and cytochrome *b* gene). Remarkably, the mutation frequency they computed for young mice ($6 \times 10^{-7}$ for rRNA mutations/bp) is very similar to the mutation frequency that we computed for our pups ($4.5 \times 10^{-7}$ mutations/bp, for rRNA, S4 Table). Another study [32], using PCR, cloning, and sequencing for comparing livers of young and old mice, found an increase in D-loop mutation frequency. However, the mutation frequency they computed was several orders of magnitude higher than ours. Yet another study [11], using ultradeep sequencing, found similarly high mutation frequencies in the whole mtDNA of mouse liver but did not detect an age-related increase [11]. These high mutation frequencies [11,32] potentially resulted from a large number of false-positive mutations, which in some instances (e.g., [11]) could have masked the signal of mutation accumulation with age. An age-related mutation accumulation was also not detected after sequencing of mtDNA from multiple wild-type mouse somatic tissues [33]. In line with our comment on germline mutations from the same study (see previous paragraph), our results suggest that this might be due to their high detection threshold MAF = 2%; we detected de novo somatic mutations with much lower MAFs ($\leq$0.69% considering all mutations and $\leq$0.31% for mutations supported by >1 DCS). Using duplex sequencing allowed us to evaluate somatic mutations at these very low allele frequencies and to show that their frequency in wild-type mice indeed increases with age, in agreement with studies of DNA polymerase subunit gamma (*Polg*) mutator mice [33].

## Mechanisms of mitochondrial mutations

The significant age-related increases in transition frequencies and in transition-to-transversion ratios we observed for both somatic and germline tissues are consistent with a major contribution of replication errors (i.e., nucleotide misincorporations) in generating mtDNA mutations. Indeed, the DNA polymerase γ responsible for mtDNA replication has a propensity for transition mutations [11,61,62]. Our results are consistent with conclusions from previous studies that also argue for replication errors as the primary source of mtDNA mutations [26,28,62–66]. Additionally, our observation of higher rates of C>T/G>A transitions at CpG than non-CpG sites, particularly for older animals, points toward a role of spontaneous deamination of methylated cytosines [54] in mtDNA mutagenesis, despite the controversial reports regarding CpG methylation in mtDNA [67–72]. The predominance of C>T/G>A mutations in a GCT context in all tissues in mothers and brain in pups (S9 Fig) is in line with a recent finding in humans showing DNA methylation predominantly at a non-CpG context, with the highest frequency in a 5′-cytosine-phosphate-thymine-3′ (CpT) and CpC dinucleotide contexts [72].

Another potentially contributing mechanism is spontaneous deamination of cytosine (C>T) and adenosine (A>G) [54,73]—also leading to transitions. In fact, the strong bias we

inferred for G>A over C>T and for T>C over A>G on the reference L-strand is consistent with a high incidence of C>T and A>G mutations occurring on the H-strand. A similar strand bias was observed in human cancer [74] and aging brain [28,29]. In mouse, this mechanism might account for the nucleotide content differences between the H- and L-strands: the H-strand is guanine and thymine rich. In addition to spontaneous deamination, a high incidence of mutations on the H-strand might be explained by its single-stranded status during the initial stages of replication (reviewed in [75]), facilitating DNA damage. Furthermore, limited repair at mtDNA might contribute to mtDNA mutagenesis [76–80]. Although we cannot completely rule out its role, oxidative damage (usually leading to transversions) does not appear to be a major player in mouse mtDNA mutagenesis [65,81].

Our results suggest that age-related mutation accumulation differs inside versus outside of the D-loop and in germline versus somatic tissues. Mouse somatic mutations appear to accumulate primarily in the D-loop with age. This finding is consistent with an elevated rate of mutation accumulation in the D-loop found in [29] but contradicts a relatively uniform mutation accumulation across the whole mtDNA molecule found in [28]—both studies examined human brain. In contrast, in the mouse germline, mtDNA mutations accumulate more evenly along the mtDNA molecule. Additionally, we observed stronger age-related frequency increases for A>G/T>C transitions in oocytes but for C>T/G>A transitions in somatic tissues. These results suggest differences in the relative contributions of various molecular mechanisms to mutations in somatic versus germline tissues and in the D-loop versus other mtDNA compartments. The latter is consistent with a recent study demonstrating a distinct mutation signature for the D-loop in humans [27,36]. Our statistical model highlights the importance of tissue, mtDNA region, mutation type, and age in determining mtDNA mutation frequency and explains approximately 24% of its variability.

## Lack of selection at de novo mtDNA mutations

Based on mutation frequencies of synonymous versus nonsynonymous variants, as well as the analysis of phastCons scores, we did not find strong evidence of selection operating on mtDNA de novo mutations. We believe this is because most of the mutations we analyzed are observed at extremely low frequencies, at which they may not have phenotypic effects on a cellular or a mitochondrial level. Therefore, our estimated mutation rates and patterns have the additional advantage of not being significantly affected by selection. Because of the very low frequency of these mutations in contrast to the high copy number of mtDNA (hundreds to around 200,000 copies, with an average of 1–10 mtDNA copies per mitochondrion in mature oocytes [16,82]), mutations in one or a few mtDNA molecules are not expected to have noticeable functional consequences for a cell compared with mutations in the nuclear genome. However, at a single-mitochondrion level, they might have a functional impact ([83], reviewed in [84]). In contrast, negative selection was demonstrated for high-allele-frequency mtDNA mutations in the mouse germline (reviewed in [85]), and it was suggested that purifying selection does not act directly on oocytes but rather on cells during postimplantation development [86]. Recent studies in humans also suggest that mtDNA variants with higher frequencies (e.g., >0.5%–1%) may be subject to negative selection in the germline [27,36] and sometimes to positive selection in somatic tissues [53].

## Random genetic drift at mouse mtDNA heteroplasmies

Analyzing inherited heteroplasmies, we found limited evidence of random genetic drift and of its increase with age in somatic tissues. Indeed, MAFs for inherited heteroplasmies in two somatic tissues correlated in mothers almost as tightly as they did in pups. Furthermore, the

normalized variance in MAF for somatic tissues was not significantly different between mothers and pups, also indicating similar amounts of drift. These observations are in contrast to the patterns observed in humans, for whom we previously observed that the correlation between mtDNA MAFs in two somatic tissues was substantially weaker for mothers than children [26]. The difference may be due to the much greater age of human mothers compared with mouse mothers.

We found a lower MAF correlation, and thus a stronger genetic drift, in mouse germ cells than in their somatic tissues—an observation consistent with the presence of an mtDNA bottleneck during mouse oogenesis [37]. Our estimate of the effective bottleneck size—84.8 segregating mtDNA units (95% CI 0.0640–462, an estimate obtained from pups)—is lower than the one previously obtained for mice, i.e., approximately 200 [37] (note that our 95% CI overlaps with 200), but higher than the ones obtained for humans, i.e., between 7 and 35 [26,36,87]. While our estimate has a wide confidence interval, it is still consistent with the view that mouse mtDNA undergoes a less pronounced germline bottleneck than human mtDNA.

Importantly, our study suggests that mtDNA experiences turnover in aging mouse oocytes. Indeed, we found that the MAF correlation between oocytes and somatic tissues is substantially stronger for pups than mothers. Thus, since drift in somatic tissues appears to be minimal, further drift might occur in maternal oocytes because of additional turnover of mtDNA in older animals. Moreover, we demonstrated a significant increase in the normalized MAF variance in oocytes of mothers versus pups. These results are in agreement with another study demonstrating an age-related increase in heteroplasmy MAF variance examined at two mtDNA sites in mouse oocytes [88] and with a recent study in humans demonstrating that mtDNA divergence between mother and offspring increases with the mother's age at childbirth [36]. Taken together, this evidence suggests that mtDNA turnover in aging oocytes might be a general phenomenon in mammals. The biological mechanism behind it is puzzling because it is usually assumed that oocytes do not divide after a female is born (except for completing meiosis right before ovulation). Transfer of mitochondria from cyst cells might play a role [55]. A decoupling of mtDNA versus nuclear DNA replication during oocyte meiotic arrest is possible and needs to be examined in future studies.

## Future directions

Our application of duplex sequencing and a unique data set comprising high-quality mtDNA sequences from two somatic tissues, single oocytes, and oocyte pools of mothers and pups from two mouse pedigrees opens avenues for future investigation of how mtDNA mutations contribute to aging and disease phenotypes [89,90]. Our study significantly advances our understanding of mouse mtDNA mutagenesis and enables a more realistic parameterization of models of mtDNA evolution. Future studies should include highly replicative tissues (e.g., intestinal epithelium) in order to further discriminate between mutation mechanisms that are dependent on or independent of replication. Such investigations should be followed by biochemical validations in vitro and in vivo. Our study is the first to use duplex sequencing for mtDNA de novo mutation analysis in germ cells and nonhuman mammalian somatic tissues. To decipher the emergence of mutations, including disease-causing ones, there is an urgent need to perform a parallel analysis in human oocytes. Additionally, to obtain a more complete understanding of mtDNA evolutionary dynamics in mammals, existing detailed studies of mtDNA de novo mutations and inherited heteroplasmies in human and mouse [26,36,56,87,88,91] should be complemented with similar studies in other mammals.

## Methods

### Ethics statement

Animals were maintained according to the Guide for the Care and Use of Laboratory Animals (Institute for Learning and Animal Research); the federal guideline followed was the Animal Welfare Act (Public Law 99–198). All animal use was reviewed and approved by the Institutional Animal Care and Use Committee (IACUC; Protocol 44472) at The Pennsylvania State University.

### Samples and their handling

Duplex sequencing was performed for somatic tissues and oocytes (single oocytes and pools of several oocytes) for mice from two independent two-generation CD-1 mouse (*Mus musculus*) pedigrees (grandparents were purchased from Charles River Labs; Fig 1; S17 Table). Somatic tissues (brain and skeletal muscle) were chosen based on their relatively high mtDNA copy number and origin in different germ layers (ectoderm for brain and mesoderm for muscle). In addition, duplex sequencing was performed for liver or heart tissue of a few selected mice to better distinguish candidate de novo mutations from transmitted heteroplasmies (S2 and S17 Tables). The ages of dams and their progeny were chosen to maximize the age difference between the two groups of mice (approximately 9 mo) at the time of oocyte collection while still maintaining the capacity to ovulate enough oocytes for subsequent analyses.

To avoid cross-sample contamination, all pre-PCR steps were performed in a designated laminar-flow hood located in a separate room. Individual samples were handled in separate low-binding tubes (Axygen). After DNA amplification, samples were handled in 96-well plate format during bead purifications with the epMotion automated liquid handling system (Eppendorf), but afterward, they were again transferred to separate tubes to minimize potential contamination among samples.

### Sample preparation, library preparation, and sequencing of somatic tissues

**DNA extraction.** Total DNA was extracted from up to 25 mg of somatic tissue using the DNeasy Blood & Tissue Kit (Qiagen) according to the manufacturer's instructions with the following modifications. After cutting the tissue samples into small pieces, they were washed with 500 μl PBS before adding 180 μl of Buffer ATL. After lysing the tissues at 56˚C for 90 min in a thermomixer, another 180 μl of ATL and 20 μl proteinase K were added to the samples that were not sufficiently lysed, and such samples were incubated in the thermomixer for another 90 min. Washes were performed with 400 μl Buffer AL and ethanol. After the wash with Buffer AL, samples were incubated at 70˚C for 10 min. Samples were then divided into two DNeasy Mini spin columns for further purification steps.

**mtDNA enrichment and enrichment estimation.** Extracted tissue DNA was enriched for mtDNA using Exonuclease V (RecBCD) (NEB), which selectively digests linear DNA while preserving the circular mtDNA [92]. This procedure has the following advantage: even if the enrichment is not 100% effective, Numts are not specifically coenriched with mtDNA, and the nuclear DNA still present in samples represents random parts of the nuclear genome. Approximately 400 ng of total DNA (such an amount of DNA was required for library preparation to obtain an mtDNA sequencing depth of >500×; however, this restricted our ability to measure high heteroplasmy frequencies within a single somatic cell or small group of cells) were incubated at 37˚C for 48 h in 60-μl reactions containing 1× NEB 4 buffer, 30 U Exonuclease V (RecBCD) (NEB), and 2 mM ATP. After the first incubation, the reaction volume was increased to 100 μl by supplementing with a further NEB 4 buffer (final concentration = 1×),

20 U Exonuclease V (RecBCD), and ATP (final concentration = 2 mM) and incubated at 37°C for another 48 h. The reactions were purified using one volume of AMPure XP beads (Beckman Coulter). mtDNA enrichment was estimated by real-time PCR, amplifying a region of the mitochondrial ND5 gene, as well as nuclear B1 elements (a repetitive element was used for the amplification of nuclear DNA to measure more efficient enrichment despite using a small amount of DNA as template): a 1:10 dilution of each sample was either amplified using primers specific for mtDNA (F-mmND5-mtDNA: CCAACAACAACGACAATCTAATTC and R-mmND5-mtDNA: TGATGGTAGTCATGGGTGGA [NC_005089.1: position 12,352–12,446]) or nuclear DNA (F-mmB1: AYGCCTTTAATCCCAGCACT and R-mmB1: CTCACTYTGW AGACCAGGCT [amplifies multiple regions in the genome]). Reactions contained 2 μl diluted DNA, 1× PowerUp SYBR Green Master Mix (Thermo-Fisher Scientific), and 0.2 μM each primer and were amplified at 95°C for 2 min, 45 cycles of 95°C for 15 sec, 56°C for 20 sec, and 72°C for 30 sec, followed by 72°C for 2 min and a melting curve analysis. A standard consisting of tissue samples that were previously enriched for mtDNA at six different efficiencies (7.2%, 10.1%, 17.2%, 21.8%, 40.0%, 46.8%, and 57.3% of mtDNA in the total sample as determined using NGS) was included in each run.

**Duplex library preparation and sequencing.** The enriched DNA was sheared to approximately 550 bp with a Covaris M220 using 50-μl reactions in a microTUBE-50 AFA Fiber Screw-Cap (Covaris) and was transferred into 200-μl low-binding tubes for library preparation (since no size-selection step is included in the protocol, a rather broad size distribution is observed in the sequenced fragments). This larger fragment size was chosen to minimize bias from shorter Numts. Duplex sequencing libraries were prepared following the previously described protocol [48] with several modifications. T-tailed adapters were prepared with sequences MWS51 and MWS55 and were purified by precipitation with two volumes of absolute ethanol and 0.5 volumes of 5 M NH$_4$OAc. End repair and A-tailing were performed according to the manufacturer's instructions with the End Prep module of the NEBNext Ultra II DNA Library Prep Kit for Illumina (NEB), followed by adapter ligation with the NEBNext Ultra II Ligation Mix and NEBNext Ligation Enhancer. Instead of the provided adapters, 25 nmole duplex adapters were used. Adapter-ligated DNA was purified twice with 0.8 volumes of AMPure XP beads (Beckman Coulter) and eluted in 15 μl of 10 mM Tris-HCl (pH 8.0). The amount of amplifiable fragments was estimated with real-time PCR as follows: 2 μl of a 1:10 diluted library aliquot was amplified in 10-μl reactions containing 1 μM NEBNext Universal PCR Primer for Illumina (NEB), 1 μM NEB_mws20 primer (GTGACTGGAGTTCAGAC GTGTGCTCTTCCGATC*T; the asterisk indicates a phosphorothioate bond), 1× Kapa HiFi HotStart Reaction Mix (Kapa Biosystems), and 1× EvaGreen Dye (Biotium). Reactions were amplified in a CFX96 Real-Time PCR machine (Bio-Rad) at 98°C for 45 sec, 35 cycles of 98°C for 15 sec, 65°C for 30 sec, and 72°C for 45 sec, followed by 72°C for 2 min and a melting curve analysis. PCR products were additionally visualized on a 1.5% agarose gel to identify samples containing residual amounts of adapter dimers. Samples in which adapter dimers were still present were repurified with 0.8 volumes AMPure XP beads and requantified before proceeding to the tag family PCR step. Tag family PCR reactions contained up to 14 μl of adapter-ligated sample. If the Cq-value in the qPCR was approximately 24, all 14 μl of the sample was used as PCR template (sequencing with approximately 7 M paired-end reads resulted in a family size of approximately 6–12, as previously suggested for duplex sequencing [48]). For lower Cq-values, samples were diluted accordingly. First, 12 cycles of linear amplification were performed (based on what is described in [93]) in 38-μl reactions: 14 μl (accordingly diluted) DNA, 1× Kapa HiFi HotStart Reaction Mix (Kapa Biosystems), 1 μM NEB_mws20 primer at 98°C for 45 sec, 12 cycles of 98°C for 15 sec, 60°C for 30 sec, and 72°C for 45 sec, followed by 72°C for 2 min. NEBNext Universal PCR Primer for Illumina (NEB) (1 μM) was

added to each reaction, followed by amplification at 98˚C for 45 sec, 9 cycles of 98˚C for 15 sec, 65˚C for 30 sec, and 72˚C for 45 sec. PCR products were purified with 0.8 volumes AMPure XP beads (Beckman Coulter) using the epMotion automated liquid handling system (Eppendorf), and different indexes were added via PCR: in 50-μl reactions, 15 μl of purified library was amplified with 1 μM NEBNext Universal PCR Primer for Illumina (NEB), 1 μM NEBNext Indexing Primer, and (NEB) 1× Kapa HiFi HotStart Reaction Mix (Kapa Biosystems) at 98˚C for 45 sec, 11–15 cycles of 98˚C for 15 sec, 65˚C for 30 sec, and 72˚C for 45 sec. The number of required cycles was estimated depending on the amount of sample used as input in the linear PCR. PCR products were purified with 0.8 volumes AMPure XP beads (Beckman Coulter) using the epMotion automated liquid handling system (Eppendorf), their quality checked with a Bioanalyzer High Sensitivity DNA Kit (Agilent), quantified with the KAPA Library Quantification Kit (Kapa Biosystems) according to manufacturer's instructions, and pooled depending on the number of reads needed per sample to obtain a family size around six. Sequencing was performed on an Illumina HiSeq 2500 platform obtaining 250-nt paired-end reads.

## Sample preparation, library preparation, and sequencing of (single) oocytes

**(Single) oocyte isolation.** Oocytes were isolated from female CD-1 mice obtained from the research colony maintained by Prof. Diaz at Penn State University. Mice were primed with 5 IU PMSG (National Hormone and Peptide Program, NIDDK) and euthanized 42–48 h later. Somatic tissues (brain, muscle, heart, and liver) were collected and immediately frozen in liquid nitrogen. Ovaries were placed in a 35-mm culture dish containing MEM-alpha (Life Technologies, Inc., Grand Island, NY) with Earles salts, 75 mg/l penicillin G, 50 mg/l streptomycin sulfate, 0.23 mM pyruvate, and 3 mg/ml crystallized lyophilized bovine serum albumin. Cumulus-oocyte complexes (COCs) were released from antral follicles by gentle puncture with a syringe and needle and washed through three dishes of media. Fully grown denuded oocytes (FGOs) were isolated from COCs by gentle pipetting to remove the cumulus cells. To prevent spontaneous resumption of meiosis, oocytes were collected in medium containing the PDE3A-specific inhibitor milrinone (10 μM). Individual oocytes were transferred to screw cap tubes, frozen in liquid nitrogen, and stored at −80˚C until library preparation.

**Oocyte lysis, mtDNA enrichment, and enrichment estimation.** The oocyte lysis protocol was modified from [94]: 4 μl of oocyte lysis buffer (50 mM Tris-HCl, pH 8.0 [Invitrogen]), 1 mM EDTA (pH 8.0) (Promega), 0.5% Polyoxyethylene (20) Sorbitan Monolaureate (OmniPur), and 0.2 mg/ml Proteinase K (IBI Scientific) were added to a single oocyte in a low-binding tube (for oocyte pools, the volume was scaled up to have about six oocytes in 4 μl) and incubated at 50˚C for 15 min, followed by an overnight incubation at 37˚C. Proteinase K was then partially inactivated by incubating at 65˚C for 30 min. In total, 1 mM ATP, 10 mM MgCl$_2$, and 10 mM Tris-HCl (pH 8.0) were added to obtain a final volume of 13 μl, and a 0.5-μl aliquot was taken, diluted 1:10, and used to (approximately) estimate the mtDNA copy number in the oocyte samples (since, occasionally, not the whole oocyte could be transferred during the lysis step—e.g., if no liquid was visible in the single oocyte tube—the measured mtDNA copy number likely underestimates the true number of mtDNA copies, as noted in S17 Table). To the remaining sample, 5 U Exonuclease V (RecBCD) (NEB) was added, followed by incubation at 37˚C for 1–4 h for single oocytes and overnight for oocyte pools. The efficiency of mtDNA enrichment was strongly dependent on the amount of media the oocytes were stored in. Higher volumes of media resulted in less efficient enrichment, independent of the incubation time. RNase A (Amresco) (10 ng) was added and incubated at 37˚C for 15 min.

Depending on the kit used for library preparation, TE buffer was added up to the final volume of 26.5 (NEBNext Ultra II FS DNA Library Prep Kit for Illumina [NEB]) or 50.5 μl (NEBNext Ultra II DNA Library Prep Kit for Illumina [NEB]). Another 0.5-μl aliquot was taken and used for mtDNA copy number estimation and mtDNA enrichment estimation. mtDNA enrichment was estimated by real-time PCR as described for tissue samples above.

**mtDNA copy number estimation.** mtDNA copy numbers, used as a quality control for our oocyte sequencing data, were estimated by qPCR amplifying a region of the mitochondrial ND5 gene (F-mmND5-mtDNA: CCAACAACAACGACAATCTAATTC and R-mmND5-mtDNA: TGATGGTAGTCATGGGTGGA [NC_005089.1: position 12,352–12,446]). One 1:10 dilution each of the directly lysed oocyte sample and mtDNA-enriched sample was used as a template. Reactions contained 2 μl diluted DNA, 1× PowerUp SYBR Green Master Mix (Thermo-Fisher Scientific), and 0.2 μM each primer and were amplified at 95˚C for 2 min, 45 cycles of 95˚C for 15 sec, 56˚C for 20 sec, and 72˚C for 30 sec, followed by 72˚C for 2 min and a melting curve analysis. A plasmid standard—the targeted ND5 region plus flanking bases (pIDTSMART-AMP + NC_005089.1: position 12,297–12,496)—was included in each run at concentrations of $10^6$, $10^5$, $10^4$, $10^3$, and $10^2$ molecules per microliter. Sample measurements were performed in duplicates, and standard measurements were performed in triplicates. Only one measurement was taken for the enriched samples.

**Duplex library preparation and sequencing.** Library preparation was performed using the NEBNext Ultra II FS DNA Library Prep Kit for Illumina (NEB) (enzymatic DNA fragmentation) for the majority of single oocytes (as described in S17 Table) and using the NEBNext Ultra II DNA Library Prep Kit for Illumina (NEB) (DNA fragmentation by ultrasonication) for oocyte pools and for some single oocytes.

With the NEBNext Ultra II DNA Library Prep Kit for Illumina, starting with the oocyte samples enriched for mtDNA, library preparation was performed as described for tissue samples, but with some modifications. In particular, instead of the adapter provided in the kit, 18 or 25 nmole duplex adapters were used for single oocytes or oocyte pools, respectively, for adapter ligation. Since Cq-values of single oocyte samples were usually higher than 23, the whole purified ligation reaction (14 μl) was used for subsequent steps, and the number of needed sequencing reads was scaled accordingly. In the indexing PCR, libraries were amplified with 10–19 cycles. The number of required cycles was estimated depending on the amount of sample used as input in the linear PCR.

Library preparation with the NEBNext Ultra II FS DNA Library Prep Kit for Illumina was performed according to manufacturer's instructions using the 26-μl oocyte sample enriched for mtDNA directly. A 5-min incubation time was used for DNA fragmentation, and the same additional changes as described for the NEBNext Ultra II DNA Library Prep Kit for Illumina were implemented.

**Quality control of fragmentation methods.** To ensure accuracy of our sequencing results, we performed three quality control experiments/analyses. First, oocyte pool G139p1 was sequenced by applying Covaris and enzymatic DNA fragmentation in parallel. Although no A>T/T>A mutations were detected in the Covaris-sheared sample, two T>A mutations were detected in the enzymatically fragmented sample. After excluding these T>A mutations (see also "Variant filtering" below), both samples displayed very similar mutation frequencies ($4.75 \times 10^{-7}$ and $3.20 \times 10^{-7}$ for Covaris- and enzymatically fragmented samples, respectively). The inherited heteroplasmies also showed similar frequencies with both methods (e.g., 26.7% and 25.5%, 12.1% and 13.4%, and 9.7% and 8.8% for sites 133 [C/A], 6570 [G/A], and 6574 [T/C] for Covaris and enzymatic fragmentation, respectively).

Second, we sequenced skeletal muscle tissue of a rhesus macaque using both fragmentation methods. Similar to the mouse sample, no AT>TA mutations were detected in the Covaris-

sheared sample, whereas two T>A mutations were detected in the enzymatically fragmented sample. After excluding these T>A mutations, mutation frequencies were very similar—$1.28 \times 10^{-6}$ and $1.39 \times 10^{-6}$ for Covaris-sheared and enzymatically fragmented samples, respectively.

Third, we analyzed the trinucleotide context of mutations separately for Covaris- and enzymatically sheared samples (S17 Fig). This analysis did not show any significant differences between the two methods—though it must be noted that we have limited power to detect significant differences here because of the small number of mutations detected for each trinucleotide context in each sample type and age group.

## Consensus read formation with barcode error correction, read mapping, and variant calling

SSCS and DCS consensus formation was performed in Galaxy [95] using the Du Novo pipeline (version 2.14) including error correction of barcode sequences (S1 Fig) [51,52]. Quality of the sequencing reads was verified using FastQC [96]. Errors in barcode sequences were corrected, allowing up to three mismatches and requiring a minimum mapping quality of 20. A minimum family size of three was required for SSCS formation, and a consensus nucleotide was called if present in at least 70% of the reads as described in [48]. The overall workflow is shown in S18 Fig.

Because of the high probability of introducing false-positive mutations (into both DNA strands) toward the ends of DNA fragments during the end-preparation step of library preparation, the first 12 nts of the consensus reads were trimmed using trimmomatic (HEADCROP) [97]. Trimmed DCSs and SSCSs were mapped to the mouse mtDNA reference genome (NC_005089.1) using BWA-MEM [98].

Both pedigrees had a mtDNA sequence length of 16,300 bp, with an insertion of an A at position 9,821.2 (TrnR) compared with the mouse mtDNA reference sequence and a T>C transition at position 9,461 (ND3 gene). Additionally, each of the two pedigrees had a fixed difference from the reference sequence: G126 had an A>T transversion at position 5,537 (in the COX1 gene), whereas G129 had a G>A transition at position 9,348 (in the COX3 gene). Those fixed differences between the two pedigrees were used to check for cross-sample contamination during library preparation. None of the sequenced samples showed evidence of cross-sample contamination.

mtDNA enrichment efficiency was estimated by calculating the percentage of sequences mapping to the mtDNA reference. In addition to mapping to the whole mouse mtDNA reference, reads were also mapped to a relinearized reference sequence (consisting of positions 14,300–15,299 and 1–2,000) to minimize mapping bias at the edges from mapping a circular genome to a linearized reference. Because an approximately 4-kb Numt with high identity to the mtDNA is present in the mouse nuclear genome [99], mapping to the whole mouse genome followed by filtering for primary alignments could not be applied (this would have led to depletion of this whole region in the analysis).

Because most mutations were observed in only one molecule, and sometimes from samples sequenced at a low depth (especially oocytes), overlapping sequences were clipped using the clipOverlap function of bamUtil [100]. This assured accurate counts of mutations in independent molecules and overall frequencies at specific sites. BAM files were filtered using the filtering tool of BAMTools [101], requiring mapping quality >20, paired reads, proper read pairing, and mapping of the mate sequence. Reads were left-aligned (Bam Left Align) to realign the positional distribution of insertions and deletions present in the sequence. Variants were called using the Naive Variant Caller (NVC) [102], using a minimum number of 1 read

needed to consider a REF/ALT, minimum base quality of 20, minimum mapping quality of 50, and ploidy of 1. Variants were annotated with the Variant Annotator tool in Galaxy [95] to retain all nucleotide substitutions that were not an N (an N is introduced in the consensus sequence if no nucleotide consensus can be formed unambiguously). From the alignment to the whole mtDNA reference, only variants found at positions 501–14,799 were used in further analyses. For the remaining positions, alignments to the relinearized reference sequence were used to call variants at positions 1–500 and 14,800–15,299 of NC_005089.1. DCSs were used for the initial variant calling and for de novo mutation analysis, whereas SSCSs were used in addition to DCSs for the analysis of inherited heteroplasmies.

## Variant filtering

If more than one variant was found in a read, a possible alignment to an alternative sequence (e.g., Numt) was evaluated with BLASTn [103]. In most cases, those fragments represented Numts, but in rare cases, they were caused by contamination with human DNA (S17 Table); those variants were excluded from further analysis. For samples with low mtDNA enrichment (<25%), all reads with a called mutation were evaluated with BLASTn to exclude potential variants arising from Numts. The analysis of the mutation pattern in single oocytes showed the presence of a high ratio of AT>TA mutations in samples prepared with the NEBNext Ultra II FS DNA Library Prep Kit for Illumina (NEB) (including an enzymatic fragmentation step) and, therefore, overall higher mutation frequencies. Single oocytes and oocyte pools prepared with the NEBNext Ultra II DNA Library Prep Kit for Illumina (NEB) (shearing of DNA with the Covaris M220 machine) did not show this AT>TA mutation prevalence. Therefore, we concluded that these represent false-positive mutations associated with the DNA fragmentation method and excluded this mutation type in enzymatically fragmented samples (this might inadvertently have led to the exclusion of some true AT>TA mutations). The higher observed ratio of transitions to transversions in single oocytes compared with oocyte pools (S7 Table) can potentially be explained by this exclusion of AT/TA mutations in the enzymatically fragmented samples, which might have also led to the exclusion of true transversion mutations, as well as by the lower number of mutations measured in oocyte pools.

## Classification and filtering of putative tissue-specific de novo mutations

Variants were analyzed for their presence in different tissues/animals/pedigrees and classified into early germline mutations (present in both somatic tissues of an animal, but absent in the germline), inherited heteroplasmy (present in [both] somatic tissues and in oocytes or in additional family members), and all other variants (putative tissue-specific de novo mutations). Putative tissue-specific de novo mutations were then further filtered based on their presence in several (unrelated) samples in the two pedigrees, as described in more detail in S1 Note. Mutations measured in >1 molecule per sample were only counted as a single mutation, assuming that they result from a single mutation event (S18 Table). All variant calling and mutation filtering steps are summarized in S19 Table.

Mutation frequency was computed as the number of mutations divided by the product of mtDNA length (16,300 bp) and mean DCS depth across samples. For instance, we computed the mutation frequency for brain in pups as 465 mutations divided by the product of mtDNA length (16,300 bp) and of the sum of mean DCS sequencing depths in brain tissue of the individual pups analyzed (i.e., 1,061,353,310 bp in total).

## Joint mixed-effects modeling

For each sample, we considered separately two different mtDNA compartments (inside and outside the D-loop) and three mutation types (transversions, A>G/T>C transitions, and C>T/G>A transitions). For each combination of mtDNA compartment and mutation type, a mutation frequency was computed as the number of observed mutations (of that type) over the number of nucleotides present in DCSs, in the considered mtDNA compartment and per sample, that can present the mutation. For example, frequencies corresponding to A>G/T>C transitions in the D-loop were computed as the number of A>G or T>C mutations observed in A and T nucleotides in the D-loop, divided by the number of A and T nucleotides in the D-loop that were present in DCSs (as proxied by the mean depth DCS times the total number of A and T nucleotides in D-loop). The data on single oocytes were combined per individual to increase the total number of nucleotides sequenced. Observations with less than 200,000 nucleotides were discarded to avoid biases due to low sequencing depth. After this preprocessing step, we retained 660 observations (out of 702).

The function glmer (from the R package lme4) was employed to fit a generalized mixed-effects linear model using a binomial family and a logit link. In particular, we considered the mutation frequency as response and included the total number of nucleotides used to compute the frequency as a weight when fitting the model. The variables age category (mothers or pups), tissue (brain, muscle, single oocytes, or oocyte pools), mtDNA compartment (inside or outside the D-loop), and mutation type (transversions, A>G/T>C transitions, or C>T/G>A transitions) were included as categorical fixed-effect predictors, whereas individual ID was included as a random effect in order to account for grouping of the observations corresponding to the same individual. We employed Wald tests (provided by the summary of glmer model) to assess the significance of each of the model coefficients (0 versus non-0), and Likelihood ratio tests (based on Chi-squared distributions) to assess both the significance of each fixed-effect categorical variable (function anova.merMod to test the full model against the reduced model obtained by removing the fixed-effect variable) and the significance of the random effect (function anova.merMod to test the full mixed-effect model against the corresponding fixed-effect model). We used the function r.squaredGLMM (from the R package MuMIn) to compute marginal and conditional pseudo-$R^2$ values (representing the variability explained by the fixed effects and by the fixed and random effects together, respectively) based on [104]. The relative contribution of a fixed-effect variable was measured through its partial pseudo-$R^2$, defined as

$$R^2_{partial} = \frac{(1 - R^2_{red}) - (1 - R^2_{full})}{1 - R^2_{red}},$$

where $R^2_{full}$ is the marginal pseudo-$R^2$ of the full model, and $R^2_{red}$ is the marginal pseudo-$R^2$ of the reduced model obtained by removing the fixed-effect variable.

The same fits and tests were employed for a mixed-effect model that, in addition to the four fixed-effects categorical variables listed above, included two-way interactions between the variable mutation type and each of the other three variables. Partial pseudo-$R^2$ values were computed as above, considering the reduced model to be the one obtained by removing the fixed-effect variable as well as the interactions involving it.

## Analysis of inherited heteroplasmies

MAFs obtained from variant calling on DCSs were used in the analysis of inherited heteroplasmies. However, if the depth of the minor allele was <5 for DCSs, MAFs obtained from SSCS

analysis were used. This allowed us to obtain a more reliable MAF in samples sequenced at low depth and for low-frequency heteroplasmies (S19 Fig).

To show the reliability of duplex sequencing in measuring low heteroplasmy frequencies, we performed control experiments by artificially mixing samples that harbor differences in their mtDNA. Because we did not have mouse samples for which the mtDNA sequence differed at several sites, we used skeletal muscle tissue from two rhesus macaques (Rh098 and Rh105), for which the mtDNA sequence differs at 14 fixed positions. Libraries were prepared as described for mouse samples, and sequencing was performed on an Illumina NextSeq platform ($150 \times 150$ paired-end reads). To mimic low-input samples, we started library preparation with approximately 200,000 mtDNA copies (which is the number of mtDNA molecules found in a single oocyte, as reported in the literature [16]). Samples were mixed at the following percentages: 10% of Rh098 and 90% of Rh105 (two replicates), 1% of Rh098 and 99% of Rh105 (two replicates), 0.2% of Rh098 and 99.8% of Rh105 (six replicates), 10% of Rh105 and 90% of Rh098 (two replicates), 1% of Rh105 and 99% of Rh098 (two replicates), and 0.2% of Rh105 and 99.8% of Rh098 (six replicates)—see S20 Table. Heteroplasmy frequencies could be reliably measured across all 14 sites and in all replicates. This demonstrates that the high variance in heteroplasmy frequency observed in oocytes does not result from the methodology employed in our study or from the use of single-cell/low-input samples but rather reflects true differences in heteroplasmy frequencies in single oocytes. By measuring a mixed frequency of 0.2%, we could also demonstrate our ability to detect such low heteroplasmy frequencies, especially when additionally analyzing SSCSs (19 out of 168 heteroplasmic sites at a mixed frequency of 0.2% could not be detected in DCSs, but all of them could be detected in SSCSs).

**mtDNA bottleneck calculation.** The size of the effective germline mtDNA bottleneck was estimated appropriately by modifying the population genetics approach developed by Millar and colleagues and Hendy and colleagues in their penguin studies [57,58]. In detail, we calculated the size of the effective mtDNA bottleneck $N$ as follows. For each individual $i$ and position $j$ we computed

$$N_{ij} = \frac{p_{ij}(1 - p_{ij})}{\sigma^2_{oocyte\ or\ pup}},$$

where $p_{ij}$ is the mean allele frequency across brain and muscle in the mother. The denominator is the mean squared deviation across $n_{ij}$ oocytes or pups for the same mouse and position, which we computed as

$$\sigma^2_{oocyteorpup} = \frac{\sum_k^{n_{ij}} \left(p_{ijk}^{oocyteorpup} - p_{ij}\right)^2}{n_{ij}},$$

where $p_{ijk}^{oocyte/pup}$ is the allele frequency for the $k^{th}$ oocyte or pup. Then, we averaged the $N_{ij}$s to obtain $N$. The 95% confidence intervals for the effective bottleneck size were generated by bootstrapping (the $N_{ij}$ values were sampled with replacement 1,000 times).

## Estimation of the rate of nonsynonymous versus synonymous mutations

We calculated the *hN/hS* ratio as described in [36,53] to test for selection in the entire protein-coding sequence (S2 Note). *hN* is the number of nonsynonymous variants per nonsynonymous sites, and *hS* is the number of synonymous variants per synonymous site. The number of nonsynonymous and synonymous sites was calculated using the Nei–Gojobori method [105].

Under neutrality, the *hN/hS* ratio is expected to be equal to one. An *hN/hS* ratio <1 is indicative of some degree of purifying selection against mutations that lead to changes in the amino acid sequence, whereas a ratio >1 can be suggestive of positive selection. The bootstrap approach [53] was used to test whether the observed *hN/hS* deviates from neutral expectations: the neutral distribution of hN/hS was generated by bootstrapping (100 replicates) from the observed mutation spectrum (the code for calculations is provided in S2 Note).

### Statistical analysis: Significance tests

One-sided permutation tests were performed to assess (1) whether the median mutation frequencies were higher in mothers (older) than in pups (younger) for each of the different tissues under analysis (100,000,000 permutations) and (2) whether the median normalized variance frequencies of heteroplasmies were higher in mothers than in pups for somatic tissues or oocytes (10,000 permutations). Here, quantities are reasonably expected to be larger in mothers.

Significance of age-related increases after combining all mutations across samples belonging to the same group (e.g., mutations in brain of pups) was assessed using the Fisher's exact test (function fisher.test from the stats package in R).

*p*-Values of all these pairwise tests were corrected for multiple comparisons (function p. adjust from the stats package in R) using the method of Benjamini, Hochberg, and Yekutieli to control the false discovery rate [106], as indicated in S4, S5, and S7–S10 Tables.

### Supporting information

**S1 Fig. Duplex sequencing—Du Novo reference-free consensus formation with barcode error correction.** With duplex sequencing, double-stranded adapters containing a random 12-nt sequence are generated for Illumina sequencing and ligated to the double-stranded, fragmented DNA of interest. Each molecule ends up with a different combination of random sequences on both ends. Adapter-ligated molecules are amplified and sequenced, producing several sequenced reads per initial DNA strand. Reads are aligned according to their random sequences, and consensus reads are formed, first for the single strands independently, SSCS, and followed by the formation of a consensus of the two strands of a DNA duplex, DCS. The duplex sequencing principle is described in detail by Schmitt and colleagues (Fig 1 in [46]). The principle of Du Novo reference-free consensus formation with barcode error correction [51,52] is shown here. Without aligning the paired-end sequencing reads to a reference sequence, reads are directly aligned according to their random tag sequences, allowing for up to three mismatches. Mismatches are corrected, and consensus sequences are formed as described above, requiring at least three paired-end reads for the formation of a SSCS and both SSCSs for the formation of a DCS. DCS, duplex consensus sequence; SSCS, single-strand consensus sequence.
(TIF)

**S2 Fig. General overview of sample and duplex library preparation, sequencing, consensus formation, and variant calling.** Total DNA was first extracted from somatic tissues and used as input for enzymatic mtDNA enrichment. Single oocytes or oocyte pools were lysed and directly used for the enrichment step. Duplex sequencing libraries were prepared and sequenced using 250-nt paired-end reads. Consensus formation was performed on Galaxy [95] using the Du Novo pipeline [51,52], and consensuses were mapped to the mouse mtDNA reference sequence (NC_005089.1) and further analyzed for variants. mtDNA, mitochondrial DNA.
(TIF)

**S3 Fig. mtDNA enrichment efficiency and sequencing depth.** (A) The efficiency of mtDNA enrichment shows a narrower distribution in somatic tissues compared with oocytes. Asterisks indicate the median of the distribution. Larger amounts of medium in which oocytes were stored reduce enrichment efficiency, resulting in oocytes with basically no, or very poor, enrichment for samples with a large amount of medium. (B) Mean DCS sequencing depth for mtDNA. A minimum mean mtDNA sequencing depth of 500× was targeted for DCS in somatic tissues (this value in reality strongly depended on the efficiency of mtDNA enrichment and is difficult to precisely estimate before tag family amplification during duplex library preparation). For single oocytes, all molecules obtained during library preparation were used as input for tag family PCR. For samples with poor enrichment only one-fourth to one-half of the library was used because the majority of the DNA represented nuclear DNA, resulting in samples with lower mtDNA sequencing depth. Asterisks indicate the median of the distribution. (C) The distribution of DCS sequencing depth across mtDNA for a typical sample (samples from mouse G131 are shown as an example). The raw data for the information depicted in this figure are available at https://github.com/makovalab-psu/mouse-duplexSeq. DCS, duplex consensus sequence; mtDNA, mitochondrial DNA.
(TIF)

**S4 Fig. Mutations introduced during somatic or germline development.** Early somatic mutations (observed in both somatic tissues). Only one of them reached MAF >1% (in brain of pup G132p1). The raw data for the information depicted in this figure are available at https://github.com/makovalab-psu/mouse-duplexSeq. MAF, minor allele frequency.
(TIF)

**S5 Fig. Mutation frequencies in germline and somatic tissues.** (A) Nucleotide substitution frequencies measured in individual oocytes of mothers and pups in the mouse pedigrees G126 and G129. Gray dots indicate a mean DCS sequencing depth <100×. Because of the low sequencing depth, mutation frequencies might be biased toward extremely low or high values (depending on the absence or presence of a mutation). (B) Mutation frequencies measured in brain, muscles, single oocytes, and oocyte pools of mothers and pups, shown at a per-individual level. Only samples sequenced at a depth of at least 100× were included (all somatic tissues; 51 of 92 single oocytes; 21 of 24 oocyte pools) to ensure accurate mutation frequency measures. For mutation frequency computation in single oocytes, the numbers of mutations and sequenced nucleotides were combined across all oocytes measured for the same mouse. Permutation test $p$-values are indicated (one-sided test based on medians; 100,000,000 permutations; corrected for multiple testing). (C) Mutation frequencies in the total mtDNA measured in brain, muscle, and oocytes (single oocytes and oocyte pools combined) of mothers and pups aggregated for all individuals of an age group. Difference between pups and mothers in each category was tested using Fisher's exact test; $^*p < 0.05$, $^{**}p < 0.01$, $^{***}p < 0.001$. The raw data for the information depicted in this figure are available at https://github.com/makovalab-psu/mouse-duplexSeq. DCS, duplex consensus sequence; mtDNA, mitochondrial DNA.
(TIF)

**S6 Fig. Mutation frequencies in the different mtDNA regions.** Mutation frequencies are shown for (A) mothers and (B) pups for the different regions along the mtDNA. Mutations frequencies in the tRNA coding regions (green) were aggregated. The raw data for the information depicted in this figure are available at https://github.com/makovalab-psu/mouse-duplexSeq. mtDNA, mitochondrial DNA.
(TIF)

**S7 Fig. Mutations in protein coding genes.** (A) Mutation frequencies within the different protein coding genes in brain, muscle, and oocytes of mothers and pup. (B) Distribution of nonsynonymous mutations (red), lost start codons (green), gained stop codons (gray), lost stop codons (yellow), and synonymous mutations (blue) within the different genes. The level of transparency represents the total number of mutations found in a gene, ranging from 1 to 47. The raw data for the information depicted in this figure are available at https://github.com/makovalab-psu/mouse-duplexSeq.
(TIF)

**S8 Fig. Mutation frequencies at CpG sites.** Significance of differences between mutation frequencies in mothers and pups was tested using Fisher's exact test; $^*$ $p < 0.05$, $^{**}$ $p < 0.01$, $^{***}$ $p<0.001$; corrected for multiple testing. The raw data for the information depicted in this figure are available at https://github.com/makovalab-psu/mouse-duplexSeq.
(TIF)

**S9 Fig. Trinucleotide context of mutations.** The trinucleotide context is shown for mutations in brain, muscle, and oocytes in mothers and pups, respectively. The nucleotide context (e.g., ACA) is only listed for one strand, but it represents the context on both strands (e.g., ACA for C>A mutations on one strand, and TGT for G>T mutations on the second strand). No significant differences in the trinucleotide context of mutations between mothers and pups was observed for brain, muscle, and oocytes ($p = 1$, $p = 1$, and $p = 1$, respectively; Pearson's chi-squared test of independence with Monte Carlo simulations). The raw data for the information depicted in this figure are available at https://github.com/makovalab-psu/mouse-duplexSeq.
(TIF)

**S10 Fig. Different types of mutations accumulate with age inside and outside the D-loop.** (A) Frequencies of different mutation types in the D-loop in pups and mothers. (B) Frequencies of different mutation types outside the D-loop in pups and mothers. Significance of differences between mothers and pups was tested using Fisher's exact test; $^*p < 0.05$, $^{**}p < 0.01$, $^{***}p < 0.001$; corrected for multiple testing. The raw data for the information depicted in this figure are available at https://github.com/makovalab-psu/mouse-duplexSeq.
(TIF)

**S11 Fig. Nucleotide substitutions accumulate asymmetrically on the two DNA strands.** We observe an asymmetric distribution of different mutation types between the L-strand (containing more cytosines than guanines) and H-strand of mtDNA, as shown previously for human brain [28]. The substitution type is shown relative to the reference sequence (representing L-strand) of mtDNA. Significance of differences between mutations on different strands was tested using Fisher's exact test; $^*p < 0.05$, $^{**}p < 0.01$, $^{***}p < 0.001$; corrected for multiple testing. The raw data for the information depicted in this figure are available at https://github.com/makovalab-psu/mouse-duplexSeq. H-strand, heavy strand; L-strand, light strand; mtDNA, mitochondrial DNA.
(TIF)

**S12 Fig. Asymmetric accumulation of mutation types with age between light and heavy strands of mtDNA.** (A) Mutation frequencies of different mutation types in the D-loop in brain, muscle, and oocytes. (B) Mutation frequencies of different mutation types outside the D-loop in brain, muscle, and oocytes. Significance of differences between different strands was tested using Fisher's exact test; $^*p < 0.05$, $^{**}p < 0.01$, $^{***}p < 0.001$; corrected for multiple testing. The raw data for the information depicted in this figure are available at https://github.

com/makovalab-psu/mouse-duplexSeq. mtDNA, mitochondrial DNA.
(TIF)

**S13 Fig. Occurrence and transmission of inherited heteroplasmies in the two pedigrees (G129 and G126).** Red numbers in italics indicate the IDs of the mothers. Circles indicate females, squares indicate males, and diamonds indicate oocytes (all single oocytes and oocyte pools were considered together). Different color fillings show the presence of different heteroplasmic sites within individuals (or all oocytes). The position of a heteroplasmic site is shown in the corresponding color when first observed within a pedigree. Positions are highlighted in yellow when first observed in mothers and in gray when first observed in pups. ID, identifier.
(TIF)

**S14 Fig. MAFs observed for inherited heteroplasmy groups separated by their occurrence in pedigrees.** MAFs are shown for heteroplasmic sites separated by tissue (brain, muscle, single oocyte, and oocyte pool) and based on their occurrence in the pedigrees. Blue: heteroplasmies shared by several mothers and their pups of a pedigree (thus were likely also present in the grandmother); green: heteroplasmies shared by a mother and some of her pups (thus likely originated in the mother or grandmother), with dark green showing heteroplasmies observed in mothers (in which they are observed first) and light green showing heteroplasmies observed in pups (in which they are inherited); red: heteroplasmies present in both somatic tissues and oocytes of one or several pups but absent from their mothers (these mutations likely originated in the germline of the mothers and were inherited by the pups). The raw data for the information depicted in this figure are available at https://github.com/makovalab-psu/mouse-duplexSeq. MAF, minor allele frequency.
(TIF)

**S15 Fig. Heteroplasmy frequencies across individual oocytes and pups.** Heteroplasmy MAFs of single oocytes plotted against the average MAF in brain and muscle of the corresponding mouse. The raw data for the information depicted in this figure are available at https://github.com/makovalab-psu/mouse-duplexSeq. MAF, minor allele frequency.
(TIF)

**S16 Fig. Relationship between normalized variance and age.** (A) Normalized variance (variance divided by $p(1 - p)$, where $p$ is the average allele frequency between somatic tissues or among single oocytes) of heteroplasmy MAFs in brain and muscle in pups and mothers. (B) Normalized variance of heteroplasmy MAFs in single oocytes of a mouse in pups and mothers. One-sided permutation test (medians; 10,000 permutations): $p = 0.478$ in (A) and $p = 0.047$ in (B). The raw data for the information depicted in this figure are available at https://github.com/makovalab-psu/mouse-duplexSeq. MAF, minor allele frequency.
(TIF)

**S17 Fig. Trinucleotide context of mutations in Covaris- and enzymatically sheared samples.** Because of the small number of mutations detected for each trinucleotide context in a sample type and age group, we do not have much power to detect differences between the used fragmentation methods. To statistically test for differences, we performed a Pearson's chi-squared test of independence with Monte Carlo simulations (we can only compute $p$-values using simulations because the assumptions of the chi-squared approximation are not met). We did not observe any significant differences in the trinucleotide context of mutations between Covaris- and enzymatically sheared samples ($p = 0.409$ and $p = 0.194$ for mothers and pups, respectively). The raw data for the information depicted in this figure are available at

https://github.com/makovalab-psu/mouse-duplexSeq.
(TIF)

**S18 Fig. Du Novo consensus formation in Galaxy [95].**
(TIF)

**S19 Fig. Correlation of heteroplasmy MAFs measured in DCSs and SSCSs.** The x- and y-axes show MAFs for inherited heteroplasmies measured from DCSs and SSCSs, respectively. (A) When only considering heteroplasmies for which the minor allele was measured in at least five DCSs, measured frequencies correlate very well in mothers as well as pups. (B) Heteroplasmies for which the minor allele was measured in less than five DCSs show a lower correlation between DCS and SSCS, likely resulting from the large confidence interval of the DCS MAFs due to the small number of molecules analyzed. Because the number of measured molecules in duplex sequencing is at least 3× higher for SSCSs, heteroplasmy MAFs from SSCSs is likely more accurate. Therefore, for these sites, MAFs measured from SSCSs were used in subsequent analyses (as indicated in S3 Table). The raw data for the information depicted in this figure are available at https://github.com/makovalab-psu/mouse-duplexSeq. DCS, duplex consensus sequence; MAF, minor allele frequency; SSCS, single-strand consensus sequence.
(TIF)

**S1 Table. Age of individual mice in the two pedigrees (G129 and G126) at the time of tissue collection.** A or B after the sample name indicates pups from two different litters.
(XLSX)

**S2 Table. Duplex sequencing summary.** Summary of the samples sequenced for different sample types, including sequencing depth, and mtDNA enrichment efficiency. mtDNA, mitochondrial DNA.
(XLSX)

**S3 Table. Tissue-specific variants found in >1 DCS (more than one mtDNA molecule in a sample; indicated as >1 DCS in the column called "notes"), early somatic mutations (found in both somatic tissues but not transmitted to the next generation or oocytes), and all inherited heteroplasmies.** DCS, duplex consensus sequence; mtDNA, mitochondrial DNA.
(XLSX)

**S4 Table. Mutation frequencies analyzed for different functional regions of mtDNA in mothers and pups.** The D-loop and noncoding region are listed separately to allow for better comparison to reported results on the D-loop region solely. The D-loop region in mouse mtDNA has a size of 877 bp compared with 934 bp considering all noncoding regions. *p*-Values were corrected for multiple testing (separately for mother versus pup comparisons and D-loop versus non–D-loop comparisons) using the method of Benjamini–Hochberg to control the false discovery rate. mtDNA, mitochondrial DNA.
(XLSX)

**S5 Table. Comparison of mutation frequencies among different mtDNA regions.** Pairwise comparisons were performed with the FET, and *p*-values were corrected for multiple testing using the method of Benjamini–Hochberg to control the false discovery rate. FET, Fisher's exact test; mtDNA, mitochondrial DNA.
(XLSX)

**S6 Table. Observed versus expected numbers of de novo mutations and inherited heteroplasmies.** (A) Somatic and germline mutations in mothers. (B) Somatic and germline

mutations in pups. (C) Inherited heteroplasmies. *p*-Values were calculated with a two-sided binomial test. The actual sequence of the two pedigrees (including one insertion compared with the reference sequence) was used. The observed numbers of variants and the numbers expected under random and neutral expectations (based on the frequency at specific sites) are shown. (D, E) One-sided binomial test to test whether mutation frequencies observed in the D-loop are significantly greater than expected by chance.
(XLSX)

**S7 Table. Frequencies of different mutation types in somatic tissues and oocytes.** *p*-Values were corrected for multiple testing (separately for mother versus pup comparisons for different mutations types, mother versus pup comparisons of Ti and Tv, and CpG versus nonCpG comparisons) using the method of Benjamini–Hochberg to control the false discovery rate. The "nt sequenced (type)" is the number of sequenced nucleotides that can effectively lead to the observed mutation type. Ti, transition; Tv, transversion.
(XLSX)

**S8 Table. Frequencies of different mutation types in somatic tissues and oocytes of mothers and pups analyzed separately for the D-loop and outside the D-loop.** *p*-Values (Fisher's exact test) were corrected for multiple testing using the method of Benjamini–Hochberg to control the false discovery rate. The "nt sequenced (type)" is the number of sequenced nucleotides that can effectively lead to the observed mutation type.
(XLSX)

**S9 Table. Analysis of strand-biased mutation accumulation.** *p*-Values for differences in mutation frequencies of the two complementary mutation types (strand bias) were calculated with FET and corrected for multiple testing using the method of Benjamini–Hochberg to control the false discovery rate. The "nt sequenced (type"' is the number of sequenced nucleotides that can effectively lead to the observed mutation type. FET, Fisher's exact test.
(XLSX)

**S10 Table. Comparison of strand-biased mutation accumulation in the D-loop versus outside the D-loop (non–D-loop).** *p*-Values for differences in mutations frequencies of the two complementary mutation types, as well as between the D-loop and outside the D-loop (non–D-loop) were calculated with FET and corrected for multiple testing using the method of Benjamini–Hochberg to control the false discovery rate (separate for strand- and D-loop/non–D-loop comparisons). The "nt sequenced (type)" is the number of sequenced nucleotides that can effectively lead to the observed mutation type. FET, Fisher's exact test.
(XLSX)

**S11 Table. Mixed-effects logistic regression model without interactions.** Fixed effects: age category (mother, pup), tissue (brain, muscle, oocyte pool, single oocyte [combined across all oocytes measured for a mouse]), mtDNA compartment (D-loop, non–D-loop), mutation type (transversion, A>G and T>C transitions, C>T and G>A transitions). Baseline for fixed effects: age category, mother; tissue, brain; mtDNA compartment, D-loop; mutations type, transversion. Random effects: mouse ID (multiple observations are related to the same individual, 36 individuals in total). Response: mutation frequency, with weights given by the number of nucleotides corresponding to each observation. Marginal psuedo-$R^2$ (represents the variance explained by the fixed effects): 22.82%. Conditional pseudo-$R^2$ (represents the variance explained by both fixed and random effects): 23.53%. Odds ratio: odds of having a mutation for the corresponding category divided by the odds of having a mutation for the baseline category (odds is defined as the probability of a nucleotide to be mutated, divided by the

probability of not being mutated). ID, identifier; mtDNA, mitochondrial DNA.
(XLSX)

**S12 Table. Mixed-effects logistic regression model with interactions.** Fixed effects: age category (mother, pup), tissue (brain, muscle, oocyte pool, single oocyte [combined across all oocytes measured for a mouse]), mtDNA compartment (D-loop, non–D-loop), mutation type (transversion, A>G and T>C transitions, C>T and G>A transitions), and interactions between mutation type (transversion, A>G and T>C transitions, C>T and G>A transitions) and each of the other variables. Baseline for fixed effects: age category, mother; tissue, brain; mtDNA compartment, D-loop; mutations type, transversion. Random effects: mouse ID (multiple observations are related to the same individual, 36 individuals in total). Response: mutation frequency, with weights given by the number of nucleotides corresponding to each observation. Marginal pseudo-$R^2$ (represents the variance explained by the fixed effects): 23.31%. Conditional pseudo-$R^2$ (represents the variance explained by both fixed and random effects): 24.02%. Interactions are significant (chi-squared test for comparison between full and reduced model): $p$-value = 1.3e−30. ID, identifier; mtDNA, mitochondrial DNA.
(XLSX)

**S13 Table. Heteroplasmic sites (inherited heteroplasmies) in the two mouse pedigrees.**
(XLSX)

**S14 Table. Heteroplasmies present either in two generations of a pedigree or in both somatic and germline tissues of an individual (i.e., inherited heteroplasmies).**
(XLSX)

**S15 Table. Calculation of correlations for inherited heteroplasmies between (A) mean MAF of heteroplasmies in brain compared with the mean MAF in muscle for pups; (B) mean MAF of heteroplasmies in brain compared with the mean MAF in muscle for mothers; (C) mean MAF of heteroplasmies in all single oocytes analyzed for a pup compared with the mean MAF of both somatic tissues of the corresponding pup; and (D) mean MAF of heteroplasmies in all single oocytes analyzed for a mother compared with the mean MAF of both somatic tissues of the corresponding mother after downsampling to equal sample sizes.** Random downsampling was performed 10x for each correlation calculation, and the mean of all obtained values was computed. (C,D) The analysis was either performed after excluding samples for which only one or two single oocytes were measured at sufficient sequencing depth (table on top) or when including all samples independent of the number of single oocytes (bottom table). MAF, minor allele frequency.
(XLSX)

**S16 Table. Estimating germline bottleneck.**
(XLSX)

**S17 Table. Overview of all sequenced libraries.** Duplex sequencing libraries for brain and muscle of 36 mice, liver of seven mice, and heart of one mouse, as well as 92 single oocytes and 24 oocyte pools. A total of five HiSeq 2500 runs were performed, labeled mm_DS01-mm_DS05. To increase the mtDNA sequencing depth, a second library was occasionally prepared for the same sample and sequenced in another run. For most single oocytes, enzymatic fragmentation, which is part of the NEB Ultra II FS DNA Library Preparation Kit, was used for DNA fragmentation (indicated with FS in column "fragmentation"); for somatic tissues, some single oocytes, and oocyte pools, shearing on a Covaris M220 machine was performed (indicated with CV in column "fragmentation"). Numbers of putative tissue-specific

mutations are shown after filtering (as described in S1 Note). mtDNA, mitochondrial DNA.
(XLSX)

**S18 Table. Complete list of de novo mutations measured in the two mouse pedigrees.** The table contains all putative de novo mutations, independent of the number of samples in which a mutation is found (indicated in the last column). Only mutations occurring in less than four samples were included in the final analysis (see S2 Note). Early somatic mutations (present in both somatic tissues, but absent in the germline) are separately listed in S3 Table.
(XLSX)

**S19 Table. Variant calling and de novo mutation filtering steps.**
(XLSX)

**S20 Table. Artificial mixtures of different heteroplasmy frequencies.** Two rhesus macaque samples differing in 14 fixed sites, and two heteroplasmic sites (one in each macaque; shown in green) were mixed at different ratios (10%, two replicates; 1%, two replicates; and 0.2%, six replicates). The mean and STDEV across all sites were calculated without the heteroplasmic sites. Nineteen out of 168 heteroplasmic sites at a mixed frequency of about 0.2% could not be detected in DCSs (MAFs) shown in red), but all of them were present in SSCSs. DCS, duplex consensus sequence MAF, minor allele frequency; SSCS, single-strand consensus sequence; STDEV, standard deviation.
(XLSX)

**S1 Note. Filtering of tissue-specific mutations based on their occurrence in the two pedigrees.**
(PDF)

**S2 Note. Selection analysis: hN/hS statistics.**
(PDF)

**S3 Note. Selection analysis: phastCons scores.**
(PDF)

**S4 Note. Mutations in regulatory D-loop regions.**
(PDF)

## Acknowledgments

We are grateful to Irene Tiemann-Boege for helpful discussions regarding the development of the library preparation protocol for single oocytes, to Kristin Eckert for reading the manuscript, and to Kristin Eckert and Suzanne Hile for their help in establishing the duplex sequencing method in the Makova laboratory.

## Author Contributions

**Conceptualization:** Francesca Chiaromonte, Kateryna D. Makova.

**Formal analysis:** Barbara Arbeithuber, Marzia A. Cremona, Nicholas Stoler, Arslan Zaidi.

**Funding acquisition:** Kateryna D. Makova.

**Investigation:** Barbara Arbeithuber, James Hester, Bonnie Higgins, Kate Anthony, Francisco J. Diaz.

**Methodology:** Barbara Arbeithuber, James Hester, Nicholas Stoler, Francisco J. Diaz.

**Project administration:** Bonnie Higgins.

**Resources:** Kateryna D. Makova.

**Software:** Nicholas Stoler.

**Supervision:** Francisco J. Diaz, Kateryna D. Makova.

**Validation:** Barbara Arbeithuber.

**Visualization:** Barbara Arbeithuber.

**Writing – original draft:** Barbara Arbeithuber, Kateryna D. Makova.

**Writing – review & editing:** Barbara Arbeithuber, Marzia A. Cremona, Francesca Chiaromonte, Francisco J. Diaz, Kateryna D. Makova.

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
