## [Editor Report · Decision Letter 0]

16 Oct 2019

Dear Dr Makova, 

Thank you for submitting your manuscript entitled "Age-related accumulation of de novo mitochondrial mutations in mammalian oocytes and somatic tissues" for consideration as a Research Article by PLOS Biology.

Your manuscript has now been evaluated by the PLOS Biology editorial staff, as well as by an academic editor with relevant expertise, and I'm writing to let you know that we would like to send your submission out for external peer review.

Please re-submit your manuscript within two working days, i.e. by Oct 18 2019 11:59PM.

Kind regards,

Roli Roberts

Senior Editor

PLOS Biology

---

## [Decision Letter · Decision Letter 1]

20 Nov 2019

Dear Dr Makova,

Thank you very much for submitting your manuscript "Age-related accumulation of de novo mitochondrial mutations in mammalian oocytes and somatic tissues" for consideration as a Research Article at PLOS Biology. Your manuscript has been evaluated by the PLOS Biology editors, an Academic Editor with relevant expertise, and by four independent reviewers.

You'll see that the reviewers are broadly positive about your study, but each has requests for textual revisions, additional analyses, and in one to two cases, new experimental data. In light of the reviews (below), we will not be able to accept the current version of the manuscript, but we would welcome resubmission of a much-revised version that takes into account the reviewers' comments. We cannot make any decision about publication until we have seen the revised manuscript and your response to the reviewers' comments. Your revised manuscript is also likely to be sent for further evaluation by the reviewers.

Your revisions should address the specific points made by each reviewer. Please submit a file detailing your responses to the editorial requests and a point-by-point response to all of the reviewers' comments that indicates the changes you have made to the manuscript. In addition to a clean copy of the manuscript, please upload a 'track-changes' version of your manuscript that specifies the edits made. This should be uploaded as a "Related" file type. You should also cite any additional relevant literature that has been published since the original submission and mention any additional citations in your response. 

Before you revise your manuscript, please review the following PLOS policy and formatting requirements checklist PDF: http://journals.plos.org/plosbiology/s/file?id=9411/plos-biology-formatting-checklist.pdf. It is helpful if you format your revision according to our requirements - should your paper subsequently be accepted, this will save time at the acceptance stage.

Please note that as a condition of publication PLOS' data policy (http://journals.plos.org/plosbiology/s/data-availability) requires that you make available all data used to draw the conclusions arrived at in your manuscript. If you have not already done so, you must include any data used in your manuscript either in appropriate repositories, within the body of the manuscript, or as supporting information (N.B. this includes any numerical values that were used to generate graphs, histograms etc.). For an example see here: http://www.plosbiology.org/article/info%3Adoi%2F10.1371%2Fjournal.pbio.1001908#s5.

For manuscripts submitted on or after 1st July 2019, we require the original, uncropped and minimally adjusted images supporting all blot and gel results reported in an article's figures or Supporting Information files. We will require these files before a manuscript can be accepted so please prepare them now, if you have not already uploaded them. Please carefully read our guidelines for how to prepare and upload this data: https://journals.plos.org/plosbiology/s/figures#loc-blot-and-gel-reporting-requirements.

Upon resubmission, the editors will assess your revision and if the editors and Academic Editor feel that the revised manuscript remains appropriate for the journal, we will send the manuscript for re-review. We aim to consult the same Academic Editor and reviewers for revised manuscripts but may consult others if needed.

We expect to receive your revised manuscript within two months. Please email us (plosbiology@plos.org) to discuss this if you have any questions or concerns, or would like to request an extension. At this stage, your manuscript remains formally under active consideration at our journal; please notify us by email if you do not wish to submit a revision and instead wish to pursue publication elsewhere, so that we may end consideration of the manuscript at PLOS Biology.

When you are ready to submit a revised version of your manuscript, please go to https://www.editorialmanager.com/pbiology/ and log in as an Author. Click the link labelled 'Submissions Needing Revision' where you will find your submission record. 

Sincerely,

Roli Roberts

Senior Editor

PLOS Biology

REVIEWERS' COMMENTS:

Reviewer #1:

The authors used duplex NGS sequencing to deeply analyze the mtDNA mutational repertoire in 20 days old pups and their 10 months old mothers (two independent pedigrees, same mouse strain - CD-1 Mus musculus). The measurements were performed in two somatic tissues/organs (brain and skeletal muscle) as well as in female germ cells (either isolated or pooled oocytes). The authors carefully assessed the distribution of identified mtDNA mutations across the mitochondrial genome in all tested samples, and used such data to compare mutation rates while taking into account age (most of the samples were females – only 9 pups were males). The authors argue that the duplex NGS, while reconstructing strand specific consensus sequences of the mtDNA, allowed dramatic reduction of sequencing errors as they focus only on variants that are in agreement between the two strands. Because of that, the authors took the liberty of recording variants below the accepted 1% threshold for the detection of mtDNA mutations. This led them to claim for the distinction between heteroplasmic and de-novo mutations (while also taking into account their recurrence in more than one tissue sample from the same individual). The authors identified over-representation of mtDNA mutations in the D-loop, and differences in the amount of mutations in mothers versus their offspring, suggesting age-related effect.

In general, I view this paper as interesting especially while employing duplex NGS to sequence DNA samples not only in bulk samples but in single (oocyte) cells. Nevertheless, there are some points that I would like to bring to the attention of the authors, which should be corrected prior to further consideration of the manuscript.

Major points:

1. In page 7 of the manuscript the authors wrote: "Across all tissues, and for either mothers or pups, aggregated mutation frequencies were notably higher in the D-loop than in protein-coding, tRNA or rRNA sequences, however they did not differ significantly among these mtDNA functional compartments (Fig. 4A, Table S5) ". While this statement refers to Fig 4 and Table S5, I carefully looked into figure 2 of the manuscript and could not find support for this statement. Figure 2 shows over representation of D-loop mutations in somatic tissues of the mothers, and in germline mutations of the pups, but certainly not in all types of samples. The authors are asked to clarify how they got to this conclusion and resolve the discrepancy between figures 2 and 4. Additionally, the rest of the paragraph relies on this statement and should be corrected or clarified accordingly.

2. Throughout the manuscript the authors discuss and compare the duplex mtDNA sequencing in tissue samples versus single oocytes. Nevertheless, the methods of isolating and sequencing DNA from single cells versus tissue samples are different and are expected to result in different sequencing read coverage, even in the mtDNA, which resides in multiple copies per cell. The authors only provided average sequence coverage (Figure S3), but did not provide sequencing reads coverage per mtDNA position per sample (which should be in the supplements). This will allow the critical reader to appreciate and compare the depth of read per mtDNA position per sample. 

3. In page 8 of the manuscript the authors wrote: "This pattern most likely reflects high mutation rates in the D-loop for these tissues, and not negative selection against de novo mutations in protein-coding regions. Indeed, we observed that such mutations exhibited nonsynonymous-to-synonymous rate ratios (hN/hS, [42]) of 1.30, 2.17, and 1.32 in brain, muscle, and oocytes of mothers, respectively, and 1.59, 1.19, and 1.69 in the same tissues of pups, all within the range of neutral expectations (Note S2)". This is a strange argument. dN/dS analysis is relevant only to protein coding genes. How can one argue against selection when there is specifically over-representation of mutations in the D-loop - a region which does not have any ORFs and hence cannot be subjected to dN/dS analysis? The over representation of mutations in the D-loop is beyond the expected portion of this region in the mtDNA (~7%), which is more logical to reflect the action of negative selection on other mtDNA regions. Secondly, selection should be analyzed and considered not only in protein coding genes, but in RNA genes while considering the conservation values (and the impact of mutations on delta G in secondary structures). Specifically, recent analysis of mtDNA mutations in the human population (Wei Wei et al Science (2019): Vol. 364, Issue 6442 - doi.org/10.1126/science.aau6520) used phastCons to assess the functional potential of mutations across the mtDNA in humans. The same approach should be performed by the authors for their mouse data. This partially relieves the dependence on dN/dS analysis; the authors are also asked to look into regulatory elements within the D-loop and compare the distribution of their identified mutations in the known regulatory elements versus the rest of the D-loop. Finally, and given the above, I feel that the authors were too fast in dismissing the possible impact of selection. After performing the requested analyses, the possible impact of selection should be discussed in the Discussion section.

4. It is unclear whether the pooled and single oocytes gave similar or different patterns in terms of mutational repertoire across the mtDNA as well as the levels of mutations per sample. The authors are asked to provide a comparison of such values.

5. The section entitled "Similar patterns of strand bias in younger vs. older animals", and its contents, needs clarification. If I understand correctly, duplex NGS enables the identification of 'true' mutations, and removal of sequencing errors, by recording mutations only in nucleotides that showed agreement between strand specific consensus sequences. I therefore am confused about the identification of strand specific mutations: are they now referring to mutations that appear in comparison to only one strand? If this is true, then it contradicts the idea presented in this manuscript, which claims to use duplex NGS to identify mtDNA mutations at high resolution. Please correct and clarify.

6. Further below in page 9, the authors wrote: "While the patterns of age accumulation inside and outside the D-loop were similar for most substitution types and tissues (Table S8, Fig. S7), C>T/G>A and A>G/T>C transitions in oocytes displayed a difference. Their frequency increased in mothers as compared to pups significantly, 2.7- and 7.1-fold, outside the D-loop, but not significantly, only 1.3- and 3.3-fold, in the D-loop. The latter observation might be explained by the small number of these mutations in the D-loop in our data set, and should be confirmed in other studies using larger sample sizes." Similar to my comment #3, such a difference in the type of mutations and their frequencies in coding versus D-loop could reflect differences in selective constraints, which should be added to the discussion of the signatures of selection. 

7. The authors report a couple of mothers who had two litters. I failed to find comparisons between the litters (per mothers), which may add some interesting information about the differences between pups and mothers.

8. In page 12 of the manuscript the authors wrote: "The resulting effective bottleneck size was estimated to be 84.8 segregating mtDNA units (95% bootstrap CI: 0.0640-462), 53.6 segregating units (95% bootstrap CI: 1.65-472), and 59.1 segregating units (95% bootstrap CI: 8.26-315), respectively (Table S16)." If I understand correctly, the authors imply that 84-59 mitochondria constitute the bottleneck of mitochondrial transfer to the maternal germline. This is very different from the published estimation of mitochondrial number in the mouse PGCs (~200 – several papers from the Chinnery group). Please explain and clarify, while referring to the relevant papers.

9. In page 13 the authors wrote: "The application of duplex sequencing to single oocytes and oocyte pools allowed us to analyze de novo mtDNA mutations directly in germ cells (i.e. without the need to observe them in the next generation) with a resolution sufficient to capture frequency increase over a period of only nine months." The authors should be more careful about such statements, as they are certainly not the first to analyze mtDNA mutations in oocytes. They need to cite and discuss their findings in light of Ma et al (2019). Deleterious mtDNA Mutations are Common in Mature Oocytes, Biology of Reproduction, ioz202, https://doi.org/10.1093/biolre/ioz202.

10. Still on page 13 the authors wrote: "Our results suggest that de novo mutations are difficult to observe with the MAF=2% detection threshold used in (Ma et al. 2018)…". In the current manuscript the authors argue for capability of duplex NGS to go below the 1% threshold, which allows identifying candidate de novo mutations. Nevertheless, to increase the confidence that the authors identified good candidates for novo mutations and not low level heteroplasmy in the germline, they should have analyzed much more than the two analyzed tissues. The authors are requested to tone down their interpretation of de novo mutations – they are only candidates for such.

11. In the Discussion sub section entitled "Lack of selection at de novo mtDNA mutations" the authors wrote: "Thus, mutations in one or a few mtDNA molecules are not expected to have noticeable functional consequences, compared to mutations in the nuclear genome. In contrast, negative selection was demonstrated for high-allele-frequency mtDNA mutations in the mouse germline [reviewed in 70]. Recent studies in humans also suggest that mtDNA variants with higher frequencies (e.g., >0.5-1%) may be subject to negative selection in the germline [18,25 (in revision)] and sometimes to positive selection in somatic tissues [42]". As mentioned in comments above, I feel that the authors rushed too fast to the lack of selection on their candidate de novo mutations. Selection is exemplified by the authors observation of preference for D-loop accumulation of mutations which is far beyond its relative portion in the mtDNA. Furthermore, yhe argument that selection works at the level of the cell and organism, is too strong: I refer the authors to a nice study by Morris et all that showed by single mitochondrial sequencing evidence for selection acting on the mtDNA even at the single mitochondrion level - Morris, J., Na, Y. J., Zhu, H., Lee, J. H., Giang, H., Ulyanova, A. V., et al. (2017). Pervasive within-mitochondrion single-nucleotide variant heteroplasmy as revealed by single-mitochondrion sequencing. Cell Rep. 21, 2706–2713. In this context, a recent review about selection in the mitochondrion should also be cited: Shtolz N and Mishmar D (2019) The Mitochondrial Genome–on Selective Constraints and Signatures at the Organism, Cell, and Single Mitochondrion Levels. Front. Ecol. Evol. 7:342. doi: 10.3389/fevo.2019.00342 

Minor points

1. In page 9 the authors wrote: "While all types of transitions displayed significant increases in mutation frequencies in mothers compared to pups (Fig. 4B; see Table S7 for mutation frequencies and Fisher’s exact test p- values), for A>G/T>C transitions the age-related frequency increase was strongest in the germline, whereas for C>T/G>A transitions – in somatic tissues.". This finding would be interesting in light of recent work that identified non-CpG methylation in the human mtDNA: Patil V., et al. (2019). Nucleic Acids Research, 47(19): 10072–10085. This citation should be added when mtDNA methylation is discussed further below in the manuscript.

2. In several places in the manuscript, the authors refer to their own manuscript which has been submitted, and is currently under revision (citation 25). This is still not an accepted manuscript, and although it has been made available to this referee, I feel that one cannot cite a paper that is not readily available to the future readers of the current manuscript. The authors should avoid such. 

3. In the section entitled " Mutation accumulation in somatic tissues" the authors should give credit to older work from the Attardi lab: Zhang J, Asin-Cayuela J, Fish J, Michikawa Y, Bonafe M, Olivieri F, Passarino G, De Benedictis G, Franceschi C, Attardi G. (2003). "Strikingly higher frequency in centenarians and twins of mtDNA mutation causing remodeling of replication origin in leukocytes", Proc Natl Acad Sci U S A.;100(3):1116-21.

Reviewer #2:

The authors have used duplex-sequencing to estimate mutational burden in mtDNA in two mouse pedigrees (total of 36 mice). They have measured mutation rate in oocyte, bulk brain and muscle. They observed increased mutation rate in older mice which confirms age effect on mutation accumulation across tissues. Moreover, the authors observed variations in patterns of mutation accumulation between germline vs soma.

The use of single cell duplex sequencing is very novel and the data set is very useful for this type of study. Although the paper can be very interesting for the readers of this journal, I believe some parts of the manuscript require major revision and re-write. Please see my main concerns below as well as some minor suggestions:

Major comments:

• Quality of data: Two different library preparation techniques have been used in the study and it is not apparent that enough has been done to investigate the errors caused by each technique. The covaris and fragmentation-based techniques seem to have been used unequally between the oocytes in mother and pups and appear to be responsible for causing the differences between oocyte pools and single oocytes seen in Figure 3. It seems logical that the mutation frequency calculated from oocyte pools and single oocytes would be more similar. The author recognizes that the fragmentation-based technique causes a large number of AT >TA mutations and deals with the problem by removing all AT >TA mutations from the study. However, the mutation frequency in the fragmentation-prepared samples still looks different from the covaris-samples. 

The use of single cell duplex sequencing is very novel – but it needs to be properly benchmarked and the authors need to demonstrate it is working correctly. I would be interested in seeing a plot of the number of mutations in each trinucleotide context for the covaris-prepared and fragmentation-prepared samples to see the differences and similarities in mutations called with these techniques. The author could also use covaris vs fragmentation as a category in their mixed-effects model to see how this contributes to mutation frequency. 

• Justification of tissue and sequencing used: the authors compared germline mutation rate vs. soma by measuring mutation rate in oocytes (single cell) vs. brain and muscle (bulk and multi cell types). Both of these selected somatic tissues are mainly post-mitotic and highly polyclonal. The author should make clear why they have selected to use brain and muscle as their selected tissues as it is not apparent. These tissues are an interesting choice as the convention in papers exploring mtDNA heteroplasmies has been to use liver. Muscle cells are multi-nucleated and it worth considering how this may affect the mutation calls. It is also not clear how many large the biopsies are – when considering MAFs it is useful to know the size of the population of cells the results came from. The study would benefit from having single cell duplex sequencing of muscle and brain. This would allow for a comparison across the tissues of single cell vs bulk sequencing and help support their results. 

• Mutation rate estimate: How the estimated de novo mutation rate from this study compares with trio data analysis? For example, how their estimate improves estimated rate, spectra in mouse pedigree by Lindsay et al., 2019 Nature communications? 

• Analysis of drift and heteroplasmies: The analysis of allele frequencies of inherited heteroplasmies in figure 5 seems to neglect some of the novel data generated in this study. By having bulk oocytes, somatic tissue and single oocytes the authors can comapre the variance of MAF at heteroplasmic sites in single oocytes against the bulk tissue MAF. Exploring mean MAF of single oocytes renders the single cell duplex sequencing less useful. 

• Mixed-effects model: The results of the mixed-effects model seem counterintuitive the results described earlier in the study. In the abstract and in the results section the authors demonstrate that the 10-month old mice have a 2-3 higher mutational frequency than the 1-month old mice implying that the mutations are age-related. However, the results of the mixed-effects model suggest that age has a very minor contribution to mutation frequency contradicting their earlier conclusions. 

• Another issue is using age as a categorical variable (as a result of the authors only collecting data at two time points). With two time points it is impossible for the authors to claim that any of the mutations are age associated, especially in the oocytes. They could be associated with parity, sexual maturity or any categorical change that occurs between 1 and 10 months. 

• Further analysis - Mutational Signatures, Hotspots, Locations: With ˜2,500 mutations called the authors have a fantastic opportunity to explore how mutations are acquired in the mitochondrial genome. In their discussion, authors mentioned: “study significantly advances our understanding of mouse mtDNA mutagenesis”, however, I do not think they have done this yet. The recent TCGA and PCAWG metanalyses as well as the Chinnery paper have demonstrated the level of detail you can explore mtDNA mutagenesis with. To make this paper more relevant to the field it might be better to explore mutagenesis using similar techniques and terminologies so that it is easier to evaluate this paper’s conclusions against the wider field eg. below:

Chinnery Paper: https://science.sciencemag.org/content/364/6442/eaau6520/tab-figures-data

TCGA Paper: https://elifesciences.org/articles/02935

PCAWG Paper: https://www.biorxiv.org/content/biorxiv/early/2017/07/09/161356.full.pdf

• The authors divide the mutation into four categories and uses these categories to explore where mutations fall in the mtDNA genome. It would be useful to see a plot showing mutation frequency across the genome to identify mutation hotspots and where mutations are occurring in genes. 

• Mention of controversy in discussion: The author mentions that this study settles a controversy surrounding age-related mutations in mtDNA in mice. However, I think this is overstating the results of studies that came before this one. The author references many studies that fail to detect mutations and one study (4) that detects mutations but fails to find correlation and one study that does find a correlation. However, the techniques used in these studies are flawed and whilst they should be refuted I do not think it is a controversial field.

Minor comments:

• The manuscript failed to cite important references. The authors have selected papers as examples to reference which, is not a best practice. All the seminal works should be cited correctly. 

• In the introduction, the authors claimed that the age related has not shown in animals with short life span, I do not think this is correct, there is at least one publication this year that looked into age effect on de novo mutation in mouse pedigree. 

• Overall the manuscript is quite lengthy and it can be more concise. Some parts of introduction and discussion is very repetitive and not clear. For example, in the introduction it is clear if the authors would like to study germline mutation rate, somatic mutation rate, or both.

• The filtering steps for identifying de novo mutations are not very clear, I suggest to generate a chart/table and include all the filtering steps. 

• Method section, copy number analysis was not even mentioned in the results neither discussion so I wonder what is the usefulness.

Reviewer #3:

Arbeithuber and colleagues utilise a highly accurate DNA sequencing approach (duplex sequencing) to investigate new mutations occurring in mitochondrial DNA (mtDNA) in mice. The study assessed oocytes and two somatic tissues (brain and muscle) from 36 mice at two different ages: pups (<1 month) and mothers (~10 months). 

The study is well structured with detailed analyses and extensive supplementary materials and code.

The study builds upon approaches uses to study mtDNA mutations in human brain, utilises sensitive sequencing methods and mouse pedigrees to provide interesting insights into mtDNA mutation accumulation in mice.

• The majority of mutations identified (1885/2015) were found in a single mtDNA molecule. All mutations in somatic tissues had low MAFs (<1%); in the germline, MAFs were as high as 7%. This may represent one of the technical differences in the study. Whilst for somatic cells the authors use 25mg of tissue, for germline they use either single cells or low numbers (<50) of cells. Can the authors comment on why this distinction was made?

• The authors identified an age-related increase in number of mutations in somatic tissue (>2x) and single oocytes, but not pooled oocytes. Can the authors comment on if there is difference in MAF between pups and mothers? Figure 5 suggests the MAFs for somatic mutations may be higher in pups? 

However, whilst DCS coverage in brain and muscle was similar in young and old mice, coverage for single oocytes was much lower in pups (median 83, as low as 1) than in mothers (median 347) (table S2). This four-fold difference in coverage may bias the likelihood of identifying mutations in the mothers’ samples. Although individual sample coverage does appear to be reported, at least one pup oocyte sample had DCS coverage as low as 1. With such low coverage, one cannot expect to find low frequency mutations. If low coverage samples are removed, is the increase in number of mutations in single oocytes still observed? 

• Utilising Duplexseq allowed the authors to assess strand bias and the authors identify a higher frequency of G>A mutations, as previously reported in human brain. Similarly, they find no evidence for oxidative damage having a major role in mtDNA damage.

• The authors all find that aging was associated with ~>3x increase in transitions in all tissues, compared to >1.2x for transversions, with C>T transitions predominating in somatic tissues. Interestingly found that T>C mutations greatly increase in germline with age. 

• Although the main focus of the study is on newly arising mutations, the authors identified 28 variants defined as ‘inherited heteroplasmies’. These were defined as being present in either: 1) both somatic and germline tissues of an individual; 2) two generations of a pedigree. 17 of the 28 were in category 1 and were not detected in the mother (Table S13). Looking at Figure S10, it appears that most of these were detected in a single pup, rather than multiple pups in the litter. Although they may represent heteroplasmy, it should be noted that mutations present in the somatic and germline tissues of a single progeny, but not parental samples are the very definition of de novo mutations. For such variants that were found in a single pup but not detected in the maternal germline, what was the DCS coverage in the maternal germline so that one can determine the ability of the assay to identify these in the maternal samples? In table S2, median oocyte DCS is ~300x, which is much lower than that of somatic cells (~2000x). Therefore, it appears that the sensitivity of the test is much lower in germ cells than in the somatic cells of the pups.

• The authors later assessed the allele frequencies of the 28 variants in tissues of pups to investigate genetic drift. In somatic tissues they found a slight increase with age. However, in the germline they detected greater variance in MAF with age, which they suggest is due to genetic drift. However, the amount of tissue analysed in germline (<100 cells) is much lower than that used for somatic tissues (up to 25mg of tissue). Thus, the bottleneck may in fact be the number of mtDNA molecules assessed in the experiment, not the biological bottleneck suggested by the authors. The authors were the first to report using Duplexseq on single cells, which is the basis for the assessment of the germline. Thus it is important to demonstrate the reliability of the approach for such low input samples. Have the authors data to demonstrate the reliability/reproducibility of the assay for single cell/low input methods?

• The large number of animals and samples analysed is commendable, although being restricted to two pedigrees and two somatic tissue types had some limitations. E.g. the authors excluded mutations present in both somatic tissues which may reflect early somatic mutations and mutations occurring more than three times at a specific site (n=148). Therefore, the study lacked the ability to identify recurrent mutations arising in independent tissues or in unrelated animals. However this study provides a good basis for such investigations in the future.

Minor comments:

• The use of the term de novo mutation in relation to somatic mutations is confusing. Typically the term de novo mutation is restricted to acquired germline mutations that are transmittable. ‘Acquired’ somatic variants or may be more appropriate and help the reader to distinguish between germline and somatic mutations. 

• Although it states in Introduction pgh3: that ‘Multi-copy mitochondrial DNA genomes (mtDNA) are transmitted’ it is important for the reader to have an understanding of the number of mitochondrial genomes per cell, and if this is known to differ among cell types. Although this is later mentioned in the discussion, it would be useful to have this information in the introduction. E.g. in the results when the median in oocytes in oocytes being 133 is mentioned, it is difficult for the reader to ascertain what proportion of mtDNA in a single cell this represents

• Introduction pgh1: ref 6 pre-dates next generation sequencing approaches. More accurate mutation rates have since been calculated using WGS. E.g. For mouse germline, see Lindsay et al. (doi: 10.1038/s41467-019-12023-w.)

• Introduction pgh1: mutation rates are ‘per basepair per generation’, not ‘per generation’

• Introduction pgh1: the authors state that conventional NGS cannot answer ‘whether their frequency increases with age’. Conventional NGS has already established this, as acknowledged by the authors in the next sentence. Therefore this should be removed. 

• Introduction pgh2: although ‘age-related frequency increases for either germline or somatic mutations have not been demonstrated unequivocally in mammals with a short lifespan’ it should be acknowledged that there have been studies with strong evidence for age-related mutation increases in much such as Lindsay et al (doi: 10.1038/s41467-019-12023-w.), Milholland et al. (doi: 10.1038/ncomms15183) etc.

Reviewer #4:

The authors report a study of mtDNA mutation in the somatic tissues of young mice, and the mothers of the pups. This allowed the authors to investigate the relative mutation frequency in the both somatic and germ cells at two defined ages, so look for variations in the mutation accumulation. The authors report a lot of intriguing findings such as the conservation of strand specific mutational biases in mammals, and that the mutation burden in the 10 month oocytes is increased relative to that in the 20 day oocytes. The report is quite interesting, and well done. The use of duplex sequencing in the question sets a new standard in accuracy and resolution (except see the comment below). 

Concerns; 

1) In the graph of the mutational frequencies given in figure 4, sup. Figures S7b, S8 and S9 show that the transition mutations general increase with age in the tissues and oocytes examined – consistent with work on aged human (reference 19). The striking exception to this pattern of C>A / G>T mutations (which also happen to be the signature of 8-oxoG mutations), where there appears to be no time-dependence in the accumulation of these mutations, near-identical frequencies between all samples in the study, and no strand dependence, as observed for the other mutation types. The fact that there is a persistent level between all samples in the study, causes me concern that there is an induction of 8-oxo-G damage in the library preparation. Damage during library preparation is something we have experienced, and it leads to the inclusion of an elevated G>T and C>T mutations. In this cause of this manuscript, it may be that the copying over damaged G’s in the earliest stages of the amplification to produce the barcoded sequences is revealing a level of 8-oxo-G damage inherent to the system. These mutations would in effect bypass the inherent proofreading of the duplex sequencing method, as the 8-oxo-G error would be amplified multiple times, leading to their inclusion in the resulting libraries (on both strands due to amplification), and passing through the error checking process. It is important that the authors test for this, as DNA damage during library preparation is well documented (https://www.ncbi.nlm.nih.gov/pubmed/23303777). I would suggest the repeating of 1 library from any tissue where there is excess DNA, but treating the preparation so that the libraries are treated with the Fpg enzyme (https://international.neb.com/products/m0240-fpg#Product%20Information) before any amplification step during the library preparation. This would allow the authors to determine whether this is occurring, and to estimate the relative amount. This would be an additional, important contribution to the development of accurate duplex sequencing. 

2) Figure 3 – the “mom” oocyte pools data appears to show a median (or possible mean - not labelled in the graph) that sits quite low in the distribution. This implies to me that there is a relatively minor increase in the “pool” mutation rate with age (which may be an artifact of the lower sequence coverage for the oocyte data) and that what is being seen is a result of clonal expansion of the existing mutations and not the accumulation of novel mutations with time. The authors should address this more deeply in the text and with more analyses, so that we can tell what dynamic is actually altered with age. Either one could be an important contribution. 

3) The paper is written in a manner that appears unaware of the fact that cyst cells import mtDNA with other organelle structures during embryogenesis (https://www.ncbi.nlm.nih.gov/pubmed/26917595). This is a rather important observation, and will affect the interpretation of things such as the genetic bottleneck, discussed in this paper. The authors should at minimum, include this information in the introduction on oocyte maturation (and its effect on discussion of genetic bottlenecks [https://www.ncbi.nlm.nih.gov/pubmed/18223651;
https://www.ncbi.nlm.nih.gov/pubmed/19029901], and frame their bottleneck discussion around these issues. There seems to be a belief out there that the Primordial Germ Cell bottleneck at <7.5dpc is “the” genetic bottleneck, and this paper could help to improve the field’s knowledge on this fact. 

Minor comments

1) One question to the experimental design – why were the ages chosen for the mothers and pups in the study? At 10 months, the females will be post-reproductive, and at 20 days, the pups will not yet be in the reproductive part of their lives. The authors should provide some rational for the ages selected – even if it was simply of experimental convenience. There are biologically more interesting time points that could have been selected, but with this sort of pioneering work, it is not a strong criticism to have selected the time point for convenience. It should just be clearly stated. 

2) Boucret et al (https://www.ncbi.nlm.nih.gov/pubmed/28938736) also sequenced human oocytes and found no mutation increase. They should also be cited alongside citation 17. 

3) 3rd paragraph of the introduction – it is clear that mitochondria have a higher substitution rate, but whether the mutation rate is higher is not clear, and dependent of the denominator used in the equation. Per generation (and per copy) – yes there are more mutation in the mitochondrial DNA compared to the nucleus. But per generation, there is more bp of mtDNA replication than the nucleus (per gene). In fact - evolutionary analyses indicated that per base –polG is the most accurate eukaryotic polymerase (https://www.ncbi.nlm.nih.gov/pubmed/21821597). It would be safer to just say “higher substitution rate” instead of “higher mutation rate”. (See also commentary on this in https://www.ncbi.nlm.nih.gov/pubmed/18288890). 

4) There is one instance of (Ma et al. 2018) as a citation instead of the [#] citation that I spotted. Please check that the citations are correct.

---

## [Decision Letter · Decision Letter 2]

27 Apr 2020

Dear Dr Makova,

Thank you for submitting your revised Research Article entitled "Age-related accumulation of de novo mitochondrial mutations in mammalian oocytes and somatic tissues" for publication in PLOS Biology. I have now obtained advice from the original reviewers and have discussed their comments with the Academic Editor. 

We're delighted to let you know that we're now editorially satisfied with your manuscript. However before we can formally accept your paper and consider it "in press", we also need to ensure that your article conforms to our guidelines. A member of our team will be in touch shortly with a set of requests. As we can't proceed until these requirements are met, your swift response will help prevent delays to publication. Please also make sure to address the Data and other Policy-related requests noted at the end of this email.

*Copyediting*

*Published Peer Review History*

*Early Version*

*Submitting Your Revision*

Sincerely,

Roli Roberts

Senior Editor

PLOS Biology

ETHICS STATEMENT:

-- Please include the full name of the IACUC/ethics committee that reviewed and approved the animal care and use protocol/permit/project license. Please also include an approval number (I cannot currently see this).

-- Please include the specific national or international regulations/guidelines to which your animal care and use protocol adhered. Please note that institutional or accreditation organization guidelines (such as AAALAC) do not meet this requirement.

-- Please include information about the form of consent (written/oral) given for research involving human participants. All research involving human participants must have been approved by the authors' Institutional Review Board (IRB) or an equivalent committee, and all clinical investigation must have been conducted according to the principles expressed in the Declaration of Helsinki.

DATA POLICY:

Many thanks for depositing your raw data in the SRA; however, we also need the numerical values that underlie the data summarized in the figures and results of your paper be made available in one of the following forms:

Regardless of the method selected, please ensure that you provide the individual numerical values that underlie the summary data displayed in the following figure panels as they are essential for readers to assess your analysis and to reproduce it: Figs 2-5 and most of the Supplementary Figures. NOTE: the numerical data provided should include all replicates AND the way in which the plotted mean and errors were derived (it should not present only the mean/average values).

REVIEWERS' COMMENTS:

Reviewer #1:

The authors adequately addressed most of my comments.

Reviewer #2:

The authors made a great improvement in their manuscript since their original submission. I am happy with their comments and I have no further questions.

Reviewer #3:

I am satisfied with the amendments made by the authors. 

Reviewer #4:

I am satisfied with the replies to my comments, and the edits and descriptions added to the manuscript. Due to time restrictions I have owing to the COVID-19 situation, I wish to state for the editors that I only focused only on my own comments, and not the broader context of all the manuscript changes in reply to all reviewers. My apologies.

---

## [Editor Report · Decision Letter 3]

27 May 2020

Dear Dr Makova,

On behalf of my colleagues and the Academic Editor, Laurence D Hurst, I am pleased to inform you that we will be delighted to publish your Research Article in PLOS Biology. 

Early Version

PRESS 

Kind regards,

Alice Musson

Publishing Editor, 

PLOS Biology

on behalf of

Roland Roberts,

Senior Editor

PLOS Biology